# A novel immunopeptidomic-based pipeline for the generation of personalized oncolytic cancer vaccines

Sara Feola[1,2,3], Jacopo Chiaro[1,2,3†], Beatriz Martins[1,2,3†], Salvatore Russo[1,2,3], Manlio Fusciello[1,2,3], Erkko Ylösmäki[1,2,3], Chiara Bonini[4], Eliana Ruggiero[4], Firas Hamdan[1,2,3], Michaela Feodoroff[1,2,3,5,6], Gabriella Antignani[1,2,3], Tapani Viitala[7], Sari Pesonen[8], Mikaela Grönholm[1,2,3,5], Rui MM Branca[9], Janne Lehtiö[9], Vincenzo Cerullo[1,2,3,5,9,10]*

[1]Drug Research Program (DRP) ImmunoViroTherapy Lab (IVT), Division of Pharmaceutical Biosciences, Faculty of Pharmacy, Viikinkaari 5E, University of Helsinki, Helsinki, Finland; [2]Helsinki Institute of Life Science (HiLIFE), Fabianinkatu 33, University of Helsinki, Helsinki, Finland; [3]Translational Immunology Program (TRIMM), Faculty of Medicine Helsinki University, postal code Haartmaninkatu 8, University of Helsinki, Helsinki, Finland; [4]Experimental Hematology Unit, Division of Immunology, Transplantation and Infectious Diseases, IRCCS San Raffaele Scientific Institute, University Vita-Salute San Raffaele, Milan, Italy; [5]Digital Precision Cancer Medicine Flagship (iCAN), University of Helsinki, Helsinki, Finland; [6]Institute for Molecular Medicine Finland, FIMM, Helsinki Institute of Life Science (HiLIFE), University of Helsinki, Helsinki, Finland; [7]Pharmaceutical Biophysics Research Group, Drug Research Program, Faculty of Pharmacy, University of Helsinki, Helsinki, Finland; [8]Valo Therapeutics Oy, Helsinki, Finland; [9]Science for Life Laboratory, Department of Oncology-Pathology, Karolinska Institutet, Solna, Sweden; [10]Department of Molecular Medicine and Medical Biotechnology, Naples University "Federico II", Pansini, Italy

*For correspondence: vincenzo.cerullo@helsinki.fi

†These authors contributed equally to this work

**Abstract** Besides the isolation and identification of major histocompatibility complex I-restricted peptides from the surface of cancer cells, one of the challenges is eliciting an effective antitumor CD8+ T-cell-mediated response as part of therapeutic cancer vaccine. Therefore, the establishment of a solid pipeline for the downstream selection of clinically relevant peptides and the subsequent creation of therapeutic cancer vaccines are of utmost importance. Indeed, the use of peptides for eliciting specific antitumor adaptive immunity is hindered by two main limitations: the efficient selection of the most optimal candidate peptides and the use of a highly immunogenic platform to combine with the peptides to induce effective tumor-specific adaptive immune responses. Here, we describe for the first time a streamlined pipeline for the generation of personalized cancer vaccines starting from the isolation and selection of the most immunogenic peptide candidates expressed on the tumor cells and ending in the generation of efficient therapeutic oncolytic cancer vaccines. This immunopeptidomics-based pipeline was carefully validated in a murine colon tumor model CT26. Specifically, we used state-of-the-art immunoprecipitation and mass spectrometric methodologies to isolate >8000 peptide targets from the CT26 tumor cell line. The selection of the target candidates was then based on two separate approaches: RNAseq analysis and HEX software. The latter is a tool previously developed by Jacopo, 2020, able to identify tumor antigens similar to pathogen antigens in order to exploit molecular mimicry and tumor pathogen cross-reactive T cells in cancer vaccine development. The generated list of candidates (26 in total) was further tested in a functional characterization assay using interferon-γ enzyme-linked immunospot (ELISpot), reducing the number

of candidates to six. These peptides were then tested in our previously described oncolytic cancer vaccine platform PeptiCRAd, a vaccine platform that combines an immunogenic oncolytic adenovirus (OAd) coated with tumor antigen peptides. In our work, PeptiCRAd was successfully used for the treatment of mice bearing CT26, controlling the primary malignant lesion and most importantly a secondary, nontreated, cancer lesion. These results confirmed the feasibility of applying the described pipeline for the selection of peptide candidates and generation of therapeutic oncolytic cancer vaccine, filling a gap in the field of cancer immunotherapy, and paving the way to translate our pipeline into human therapeutic approach.

## Editor's evaluation

The reviewers find that the findings presented in this study are interesting and have an immediate impact on immune oncology.

## Introduction

The ligandome describes the peptide composition bound to the major histocompatibility complex (MHC) I and II presented on the cellular surface (*Freudenmann et al., 2018*). Once being identified as targets by the immune system, the peptides in the MHC-I are the contact point between cytotoxic CD8+ T cells and the tumor cells. Thus, the knowledge of those peptides is a key point in designing therapeutic cancer vaccines to generate and stimulate specific antitumor adaptive immune responses. Moreover, the interest in identifying and exploiting these targets gained momentum following the breakthrough of the immune checkpoint inhibitors (ICIs) as it became clear that ICI treatment can unleash the specific antitumor T-cell responses against these immunogenic candidate targets.(*Bassani-Sternberg et al., 2016*). Indeed, the ICI therapy activates a preexisting antitumor immune response with immune cell infiltration in the cancer lesions, defined as 'hot' tumors; instead, tumors not infiltrated with immune cells are called 'cold.' As a result, the response rate to the ICI therapy can vary from 40% to 70% to 10% to 25% either due to the lack of immune cell infiltration into the tumor or other immunosuppressive mechanisms in the tumor microenvironment (TME) (*Schoenfeld and Hellmann, 2020*; *Wieder et al., 2018*). Currently, there is an urgent need to find a way to turn 'cold' tumors to 'hot' ones, making the ICI therapies more effective. In this context, the development of effective peptide-based cancer vaccines for therapeutic approaches is facing two main challenges: the criteria to select peptides able to elicit an immune response and the use of an adjuvant to increase the antitumor immune response of the immunizing peptides. In this article, to overcome these issues, we have developed a pipeline that covers the diverse developmental stages of therapeutic cancer vaccines, moving from the isolation of the MHC-I-restricted tumor peptides, to the selection and screening of target candidates until the generation of an oncolytic cancer vaccine. First, we selected the known murine immunogenic tumor model CT26, allowing the study of the antitumor response (*Lechner et al., 2013*). We investigated the MHC-I antigen landscape of CT26 applying state-of-the-art immunopeptidome and mass spectrometric methodologies. The immunopeptidome profile was carefully analyzed and found to be qualitatively in line with already published dataset; the result list of peptides was then investigated through two approaches: RNAseq and HEX software. The latter is a tool that identifies tumor antigens similar to pathogen antigens, exploiting the cross-mimicry and cross-reactive T cells for clinical applications (*Jacopo, 2020*). The peptides derived from those analyses were then investigated in vivo by pre-immunizing mice with the adjuvant poly:(IC) and the peptides; the splenocytes were then harvested and functional characterization was performed by interferon-γ enzyme-linked immunospot (ELISpot), deconvoluting the single-peptide immunogenicity. For the last part of our pipeline, after the functional characterization, the selected peptides were used to generate an oncolytic cancer vaccine. To take full advantage of viral immunogenicity to induce a specific antitumor T-cell response, we used our previously developed platform, PeptiCRAd, based on an oncolytic adenovirus (OAd) coated with immunogenic tumor antigen peptides (*Capasso et al., 2016*; *Feola et al., 2018*). The peptide candidates in this study were tested in our PeptiCRAd platform, which here consisted of a conditionally replicating OAd armed with two immune-activating ligands, the ligand for cluster of differentiation 40 (CD40L) and the ligand for tumor necrosis factor receptor superfamily member 4 (OX40L), named VALO-mD901 (*Ylösmäki et al., 2021*). Intratumoral

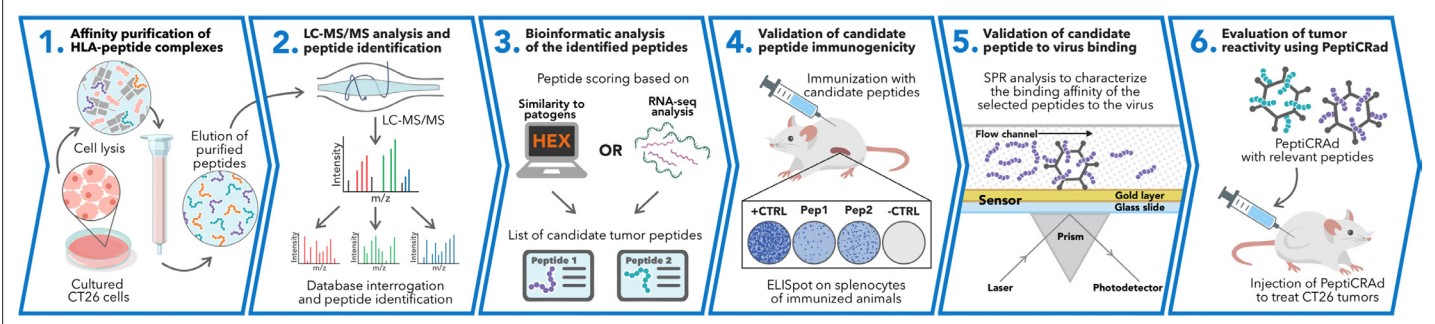

**Figure 1.** Schematic of the proposed immunopetidomic-based pipeline. Major histocompatibility complex (MHC) I peptides are immunopurified from the surface of tumor cells (**Step 1**). Next, the peptides are analyzed by mass spectrometry (**Step 2**) and the generated list of peptides is investigated with two main approaches: RNAseq analysis and HEX software (**Step 3**). The selected peptides then go through a functional characterization for their immunogenicity profile in vivo through enzyme-linked immunospot (ELISpot) assay (**Step 4**) and the best candidates are poly-lysine-modified and analyzed by surface plasmon resonance (SPR) for their binding affinity to the oncolytic adenovirus (OAd) (**Step 5**). Finally, the peptides are used to decorate OAd to generate therapeutic cancer vaccine (PeptiCRAd) and tested in tumor-bearing mice (**Step 6**).

The online version of this article includes the following figure supplement(s) for figure 1:

**Figure supplement 1.** Flow cytometry analysis of H2K$^d$ expression level in the colon tumor model CT26.

administration of PeptiCRAd coated with the peptides selected based on our pipeline controlled the tumor growth in CT26 tumor-bearing mice. Additionally, we observed that the specific antitumor immune activation generated in the primary tumor could be extended to a second tumor lesion in a phenomenon known as 'abscopal effect.' Thus, we developed and validated a pipeline moving from the isolation of the peptides to the selection of the target candidates until the combination of these in our PeptiCRAd platform, showing the efficacy in a preclinical model of colon cancer on to the primary tumor and distant lesions.

To the best of our knowledge, the described pipeline covers for the first time all the stages of a personalized therapeutic cancer vaccine development, starting from the isolation of MHC-I-restricted peptides derived from the primary tumor to their analysis in silico and in vivo to identify the best target candidates. Finally, an OAd was coated with these peptides to generate an effective therapeutic cancer vaccine. The pipeline can be translated to personalized cancer treatment in relevant clinical application as the OAd can be easily coated with the unique repertoire of patient-specific tumor peptides profile, a prerequisite for personalized therapy.

## Results

### Immunopeptidomic analysis reveals the MHC-I profile in a preclinical model of colon cancer

The identification and selection of candidate targets followed by the generation of therapeutic cancer vaccines is a scattered rather than a complete workflow. This drawback prompted us to develop a comprehensive pipeline that could cover the major steps in the process. First, we aimed to directly isolate MHC-I-restricted peptides from the tumor surface as they are the key contact points between the tumor cells and the cytotoxic CD8+ T cells (*Figure 1*, Step 1). Next, the peptides were analyzed by mass spectrometry (*Figure 1*, Step 2) and the generated list of peptides was investigated with two independent approaches: RNAseq analysis and HEX software (*Figure 1*, Step 3). The selected peptides were then functionally characterized for their immunogenicity profile in vivo by ELIspot (*Figure 1*, Step 4) and the best candidates were modified to contain polyK attachment moiety and were analyzed by surface plasmon resonance (SPR) for their binding affinity to the OAd (*Figure 1*, Step 5). Finally, the peptides were used in our PeptiCRAd cancer vaccine platform (*Figure 1*, Step 6). As we sought to investigate whether the proposed pipeline could be applied for the development of therapeutic cancer vaccines, we selected the known immunogenic model CT26 (*Lechner et al., 2013*) that expresses high surface level of MHC-I as shown in our flow cytometry data (*Figure 1—figure supplement 1*). We immunopurified MHC-I-restricted peptides and analyzed the eluted peptides by tandem mass spectrometry. By using the murine reference proteome and applying a false discovery

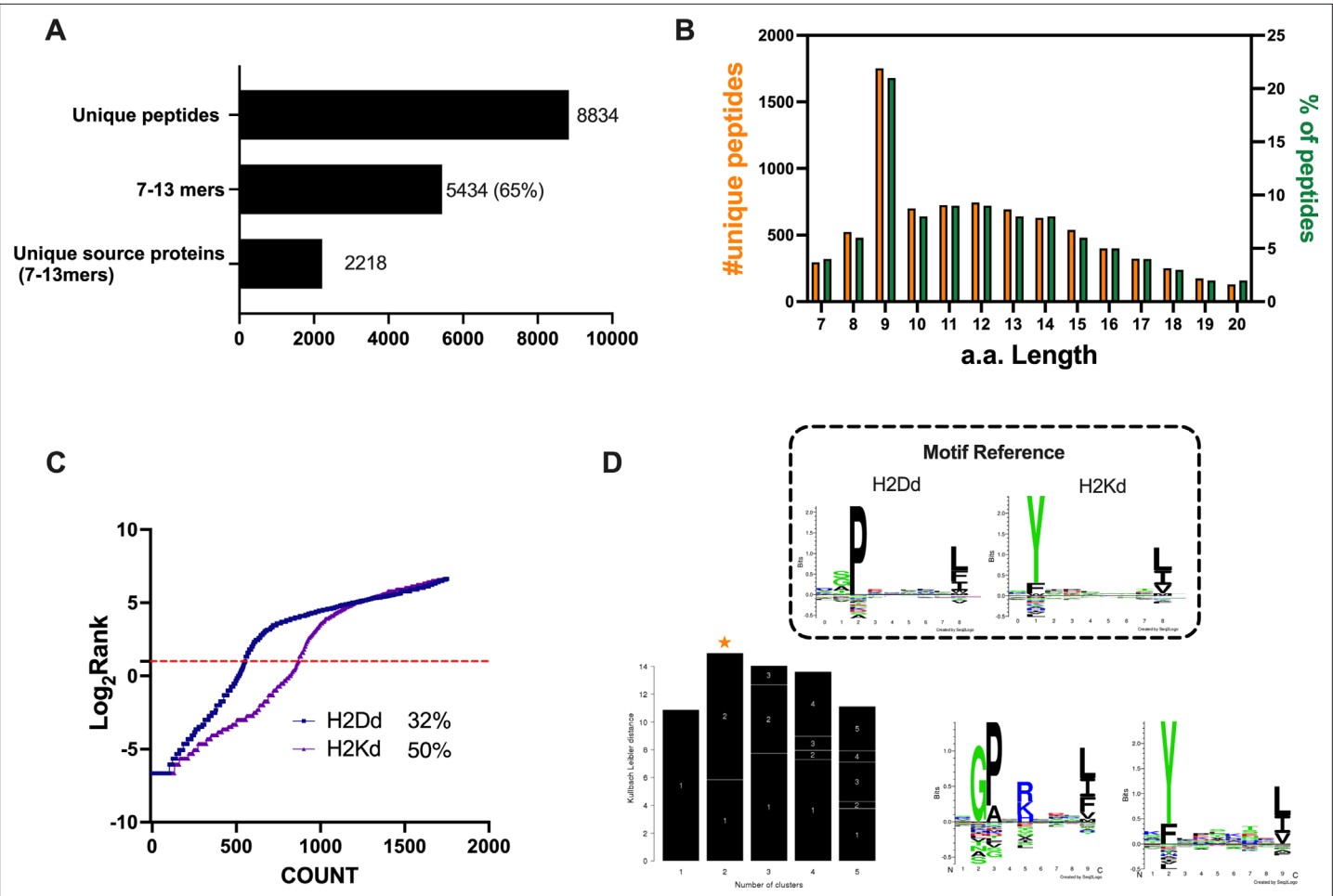

**Figure 2.** Properties of the peptides eluted from the CT26 tumor model. (**A**) Unique peptides, 7–13 specimen, and their respective source proteins are reported as finite number and depicted as bar plots. (**B**) Overall peptides' amino acid length distribution is shown as function of number (left y-axis) and percentage of occurrence (right y-axis). (**C**) The eluted 9mers were analyzed in regard to their binding affinity to H2K$^d$ and H2D$^d$. Binders and not binders were defined in NetMHCpan 4.0 Server (applied rank 2%). (**D**) Major histocompatibility complex (MHC) I consensus binding motifs. The consensus binding motifs among the eluted 9mers peptides were deconvoluted through Gibbs clustering analysis. The reference motif (according to NetMHCpan motif viewer) is depicted in the upper square. The clusters with the optimal fitness (higher KLD values, orange star) are shown, and the sequence logo is represented.

rate (FDR) threshold of 5% for peptide identification, a total of 8834 unique peptides were identified (*Figure 2A*). In order to assess the overall performance of the immunopurification of the MHC-I-restricted peptides, we carefully investigated the presence of contaminants in the immunopurified peptides. Among those, the 7-13mers accounted for 5434 peptides (65% of the total eluted peptides) derived from 2218 unique source proteins (*Figure 2A*). The peptides showed the typical amino acid length distribution profile with the 9mers as the most enriched fraction, representing 21% of the total amount of peptides (*Figure 2B*). Next, the analysis of binding affinity to MHC-I showed that 81% (1413 of 1752) of 9mers were binders either for H2K$^d$ or H2D$^d$ (according to NetMHC4.0, applied rank <2%) with 62% of the binders showing preference for the H2K$^d$ allele (*Figure 2C*). Moreover, Gibbs analysis was used to deconvolute the consensus binding motifs of respective MHC-I alleles from the eluted 9mer peptides; these clustered in two distinct groups, with a preference for reduced amino acid complexity for residues at positions P2 and P9, matching remarkably well with the known motifs for H2K$^d$ and H2D$^d$ (*Figure 2D*). Overall, the analysis outcome was similar to published dataset (*Schuster et al., 2018*) (amino acidic length distribution, Gibbs clustering profile, amount of binders), confirming the good quality of the ligandome landscape identified.

Then, we aimed to investigate whether the MHC-I source proteins identified among the binders (9mers) were attributable to a specific biological process. Indeed, MHC-I peptides are predominantly

derived from cytosolic/nuclear proteins, which normally do not intersect the endocytic compartment and are mainly involved in maintaining the structure of the cell (cell proliferation, differentiation, signaling, translation) (*Adamopoulou et al., 2013*). To this end, we performed a Gene Ontology (GO) enrichment analysis. As expected, the biological process highlighted the enrichment in pathways that comprise regulation of chromosome organization, DNA repair, ribosome biogenesis, RNA splicing, DNA-protein interactions, and cytoskeleton organization (*Figure 3A*). Moreover, the linkage between the genes and the biological process depicted an overrepresentation of epigenetic regulators (e.g., histones, DNMT1) (*Figure 3B*, *Figure 3—figure supplement 1*), in line with preceding reports in the literature (*Löffler et al., 2018*). The cellular component (CC) and molecular functions (MFs) confirmed the nature of the source proteins, showing an enrichment, for instance, in nucleosome and chaperone proteins, respectively; these are well-known sources of MHC-I ligands (*Figure 3—figure supplement 2*).

Overall, these analyses assessed and demonstrated the reliability of the generated ligandome dataset, confirming the robustness of the peptides' list as true ligands and allowing us to proceed further with the downstream applications.

## In silico prediction of candidate targets based on RNAseq analysis and similarity to pathogen antigens

We carefully examined the list of generated peptides to check for the presence of contaminants, and based on the aforementioned analysis, the eluted peptides resembled the MHC-I ligandome landscape. As we sought to generate and develop an effective therapeutic cancer vaccine, we next moved to select the best peptide candidates that could elicit a strong adaptive immune response. However, the criteria for selecting and narrowing down the number of peptide targets are still challenging for the field, usually involving laborious and time-consuming approaches and remaining therefore a critical question to address (*Laumont et al., 2018*). To overcome this limitation, we analyzed the list of peptides adopting two parallel approaches. The first one is based on the RNA expression level of the source proteins of the MHC-I ligands. With this mind, we first identified the transcripts (and thus the corresponding source proteins) overrepresented in CT26 tumor cell line compared to normal cells. The RNAseq profile of the syngeneic medullary thymic epithelial cells (mTECs) and the colon Balb/c was used as normal control. Thus, we analyzed the differential gene expression (DESeq) profile between the CT26 and mTEC (*Figure 4A*) and CT26 and colon (*Figure 4B*) (standard cutoff values of fold change 1.5 and a padj-value of 0.05, red square); then, we searched the source proteins of the 9mers ligands derived from our previously generated ligandome dataset (red dots in *Figure 4A and B*) in the DESeq data for each expression profile analysis. In order to identify tumor-associated antigens (TAAs), we selected the liagandome source proteins for which the corresponding transcripts were overexpressed in both DESeq analyses (*Figure 4A and B*, red dots within the red square). Finally, we further investigated the chosen candidates, prioritizing the peptides with source proteins that have transcript-level high fold change for both DESeq analyses and simultaneously a strong binding affinity for both H2K$^d$ and H2D$^d$ allotypes (cutoff values -log$_{10}$ 0.5 H_Average ranks and third quartile of average fold change; *Figure 4C*), generating the final list of candidates (*Supplementary file 1*).

The second approach consisted of using the HEX software to inspect the sequences of MHC-I ligands for similarity to antigens from pathogen. First, the software prioritized the peptides that were concurrent strong binders (cutoff IC$_{50}$ range 50–500 nM according to NetMHC4.0) and that showed higher weighted alignment score (cutoff 0.8–1 normalized weighted alignment score). The latter focuses on the peptide's similarity in the area of interaction that most likely will engage the T-cell receptor (TCR) of CD8+ T cells in order to mediate immune response (*Figure 4D*); the resultant peptides are then further categorized based on their overall percentage of identity to various pathogen antigens and IC$_{50}$ binding affinity score (*Figure 4D*). The ultimate output consisted of 13 peptides with their counterpart pathogen peptides (*Supplementary file 2*). Thus, the list of candidates derived from RNAseq analysis and HEX software accounted for 26 peptides. The peptides where then functionally characterized in in vivo setting. To determine the peptide immunogenicity, mice were pre-immunized with subcutaneous injection of each peptide in the presence of the adjuvant poly(I:C) and a group of mice was injected either with poly(I:C) alone or saline as control as well. The splenocytes from those mice were harvested and tested for IFN-γ production upon specific stimuli in an ELISpot assay, according to the peptide identification number presented in *Supplementary file*

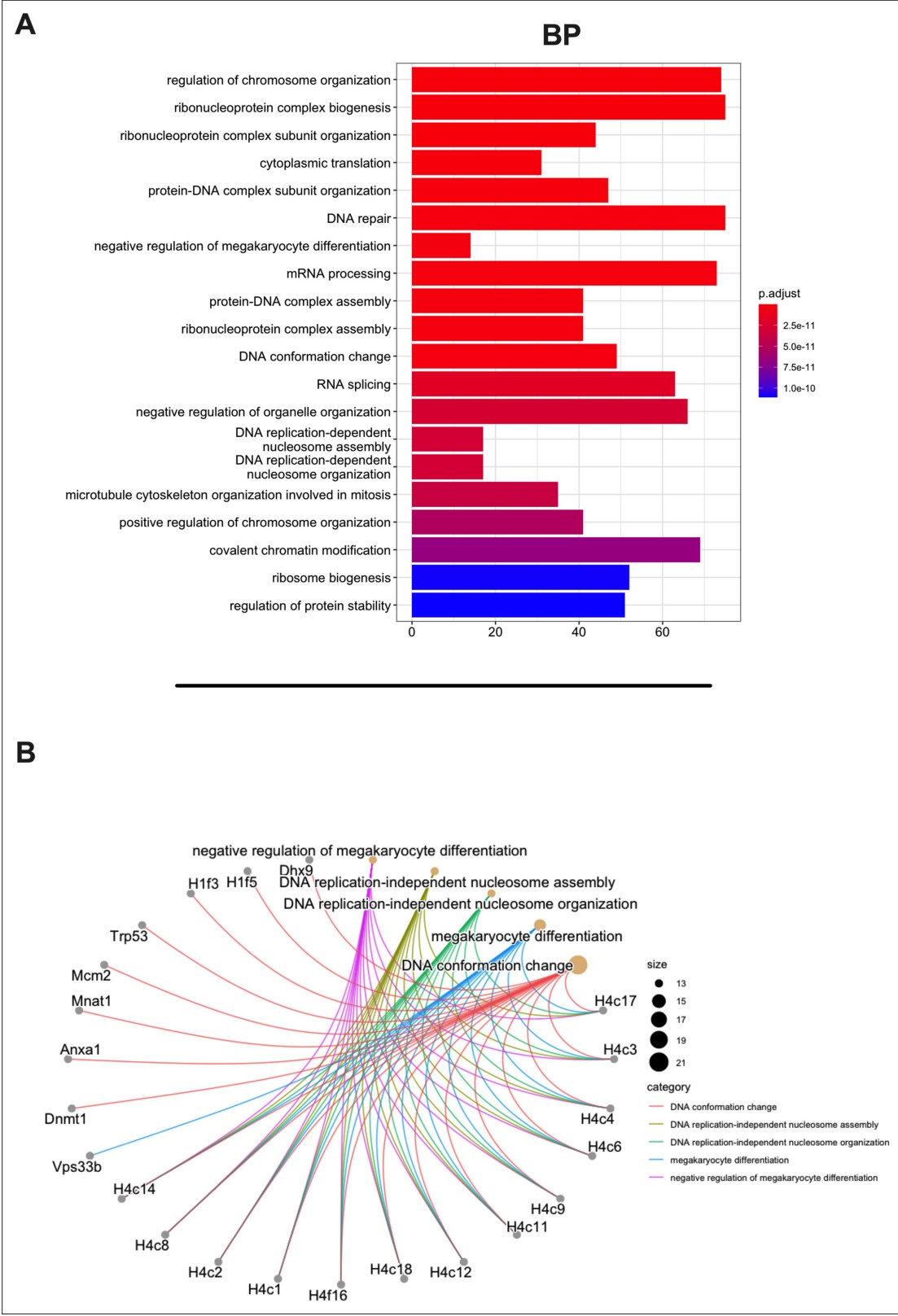

**Figure 3.** Gene Ontology (GO) enrichment analysis of the source proteins. (**A**) GO enrichment was evaluated by biological process (BP); adjusted p-values of the first 20 statically relevant BPs are depicted as color gradient, and the respective number of genes is shown as bar plots. (**B**) Genes and BP linkages are summarized in a cnetplot graph. Each color line represents a different BP category, and the bubble size symbolizes the number of genes.

*Figure 3 continued on next page*

*Figure 3 continued*

The online version of this article includes the following figure supplement(s) for figure 3:

**Figure supplement 1.** Heatmap of the Gene Ontology (GO) enrichment results is displayed.

**Figure supplement 2.** Gene Ontology (GO) analysis is reported.

3. Our data showed that six peptides induced higher frequencies of T-cell-specific response (*Figure 5*, red squares) defined as the average of the number of spots above the threshold of at least 100 (peptide 4) that is at 10-fold change compared to the control groups. Next, the six peptides selected in the ELISpot assay were modified to contain poly-lysine attachment moiety (polyK-peptides) at the N-terminus to increase the net charge at pH 7 (*Supplementary file 4*) and tested for their electrostatic interaction with the OAd; to this end, (3-aminopropyl) triethoxysilane (APTES) silica $SiO_2$ sensors were first coated with the VALO-mD901 and then 100 µM of polyK- peptides were injected into the SPR system. Peptide 7 is gp70$_{423–431}$ (AH1-5), a known immunodominant antigen of CT26 derived from a

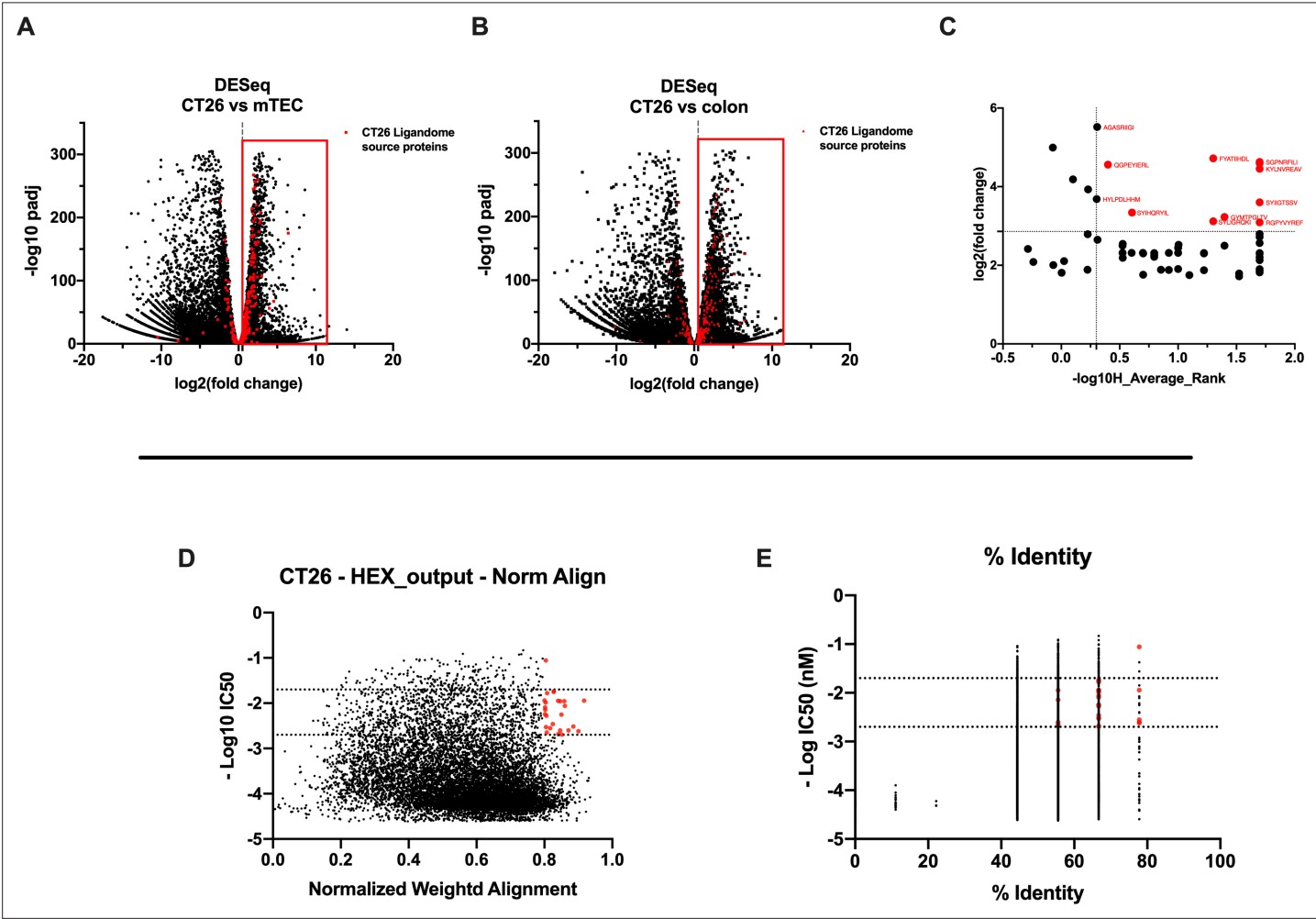

**Figure 4.** Differential expression and HEX analysis for the major histocompatibility complex (MHC) I ligand candidates. (**A, B**) Differential gene expression profile (DESeq) in CT26 versus medullary thymic epithelial cell (mTEC) (**A**) and CT26 versus healthy Balb/c colon (**B**) is depicted as volcano plot of -$\log_{10}$ of p-adj-values versus $\log_2$ ratio (fold change). The source proteins of MHC-I ligands from our dataset are marked in red, and the differential expression is considered significant for a fold change of 1.5 and a padj-value of 0.05 (red square). (**C**) Scatter plot comparing the fold change of the source proteins found statistically overexpressed in both DESeq analysis and the average binding affinity score for both H2K$^d$ and H2D$^d$ allotypes. The values were considered significant for >-$\log_{10}$ 0.5 H_average ranks and the third quartile of average fold change (red marked). (**D, E**) The peptides were stratified based on their binding affinity expressed as -$\log_{10}$ and on the weighted score to prioritize similarity between more central amino acids in the peptide (**D**) or on the percentage of similarity to viral peptides (**E**). Binding affinity <50 nM and weighted score and similarity >0.8 were considered as the threshold to select tumor peptides similar to viral epitopes.

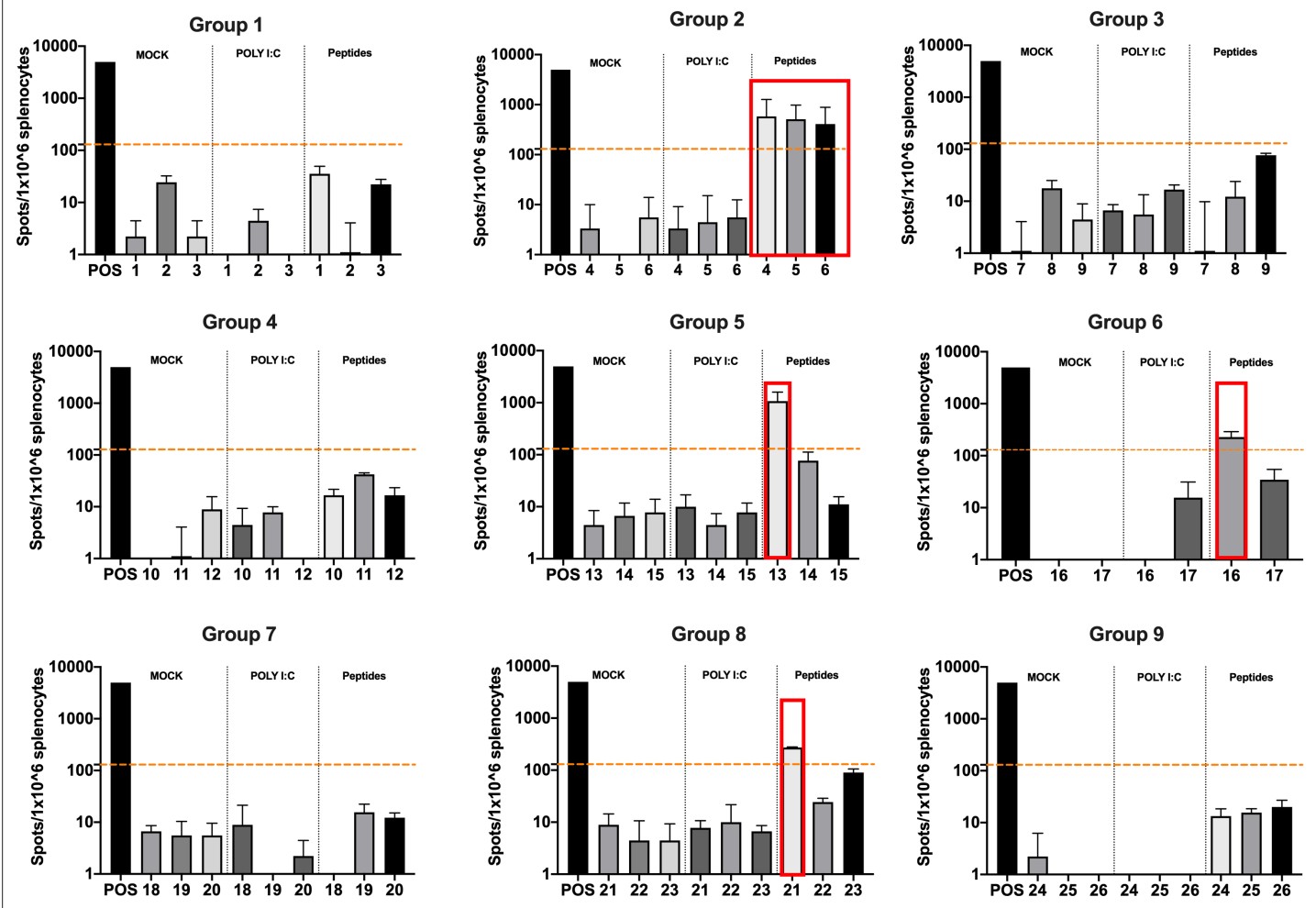

**Figure 5.** Functional characterization of the peptide candidates. Enzyme-linked immunospot (ELISpot) IFN-γ analysis was performed on splenocytes harvested from mice pre-immunized with poly(I:C) and the peptide candidates. The figure shows the stimuli conditions and the treatment groups. The frequencies of antitumor T-cell responses are depicted as peptides-specific reaction per $1 \times 10^6$ splenocytes. The average of the number of spots above 100 (i.e., 10-fold change compared to the control groups' signal, orange dashed line) is defined as the inclusion criteria to select the peptides (red square).

viral envelope glycoprotein encoded in the genome, and it was analyzed as well to exploit it as control in downstream animal experiment. The interactions of OAd with the peptides were measured at equilibrium (MAX) and dissociation (MIN) points (*Figure 6A*). At equilibrium point, all peptides showed interactions with OAd (*Figure 6B and C*). However, at dissociation stage, peptides 1, 2, 6, and 7 reached the highest number of peptides retained for viral particle (VP).

In summary, the in vitro and in vivo validation and characterization guided the selection of candidate peptides to be used with our PeptiCRAd technology to elicit antitumor T-cell response.

## PeptiCRAd platform induces systemic antitumor immune response controlling the tumor growth of distant untreated cancer lesion in a murine model of colon carcinoma

By applying RNAseq and HEX software followed by an in vivo functional characterization, we identified six peptides to be tested (*Supplementary file 4*, peptides 1–6) in the PeptiCRAd cancer vaccine platform. The adenovirus used in the PeptiCRAd platform was VALO-mD901, genetically modified to express murine OX40L and CD40L and previously shown to elicit tumor growth control and systemic antitumor response in a murine model of melanoma (*Ylösmäki et al., 2021*). Therefore, immunocompetent Balb/c mice were subcutaneously injected with the syngeneic CT26 tumor cells in the left and

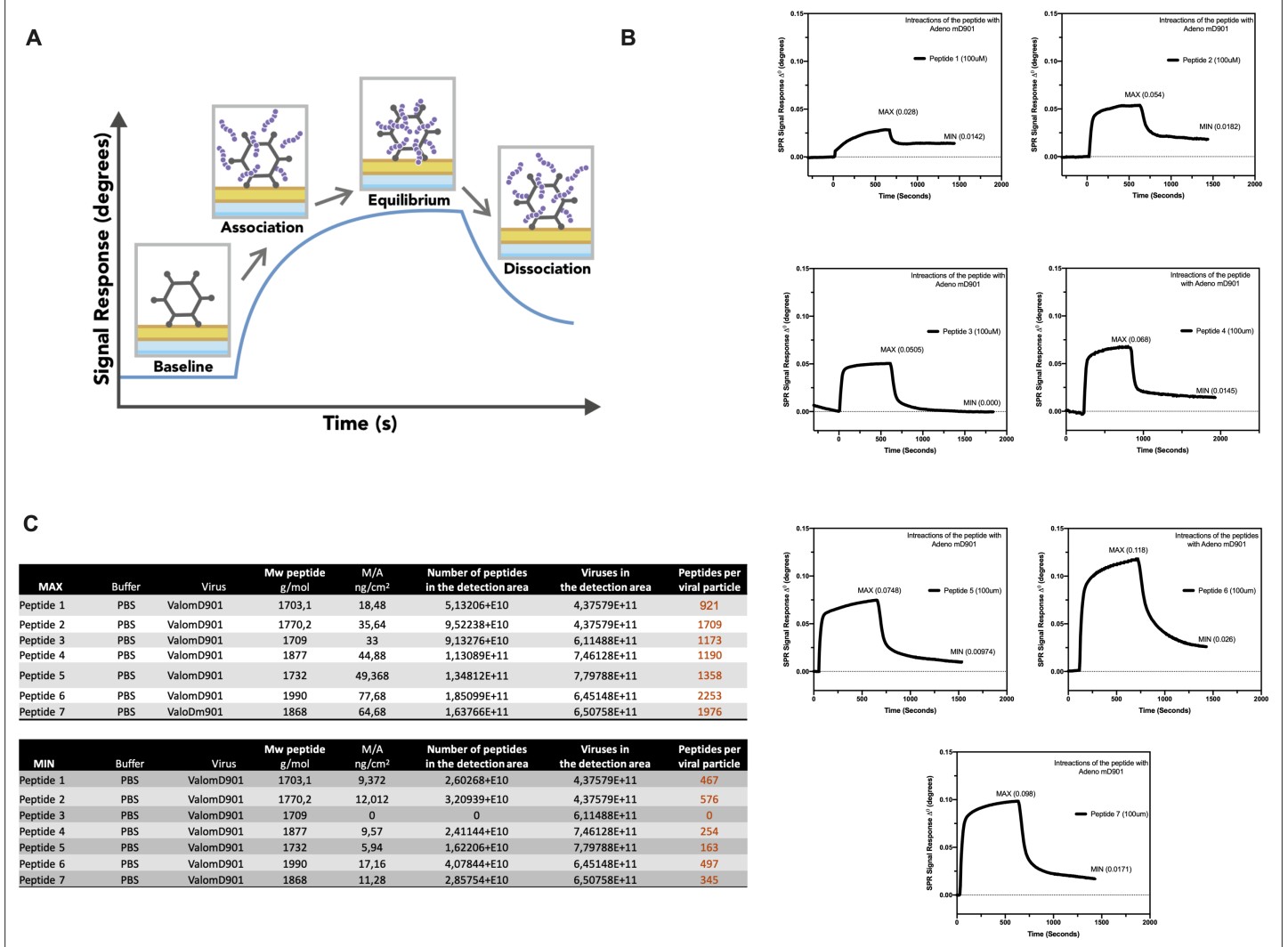

**Figure 6.** Surface plasmon resonance (SPR) analysis of the peptide/oncolytic adenovirus (OAd) interaction. (**A**) An overview of the SPR analysis principle is depicted. (**B**) SPR analysis of the interaction between the poly-lysine-modified peptides and OAd is shown as signal response degree and time (seconds). For each peptide, the maximum interaction (MAX, equilibrium) and minimum (MIN, dissociation) peak are reported. (**C**) For each peptide and both equilibrium and dissociation stage, the number of peptides per viral particle has been determined.

right flanks (day 0, **Figure 7A**). When the tumors were established (day 7, **Figure 7A**), VALO-mD901 was coated with a pair of each polyK-peptide in our list (PeptiCRAd1, PeptiCRAd2, PeptiCRAd3, **Supplementary file 5**) and injected intratumorally only in the right tumor. PeptiCRAd4 consisted of VALO-mD901 coated with gp70$_{423-431}$ (AH1-5); mock and VALO-mD901 groups were used as controls. PeptiCRAd1 and PeptiCRAd2 improved tumor growth control as well as VALO-mD901 in the injected lesions (**Figure 7B**, right panel) as depicted also in the single-tumor growth curves per each mouse per each treatment group (**Figure 7—figure supplement 1**). In addition, PeptiCRAd1 (PC1) showed a clear trend towards an improved antitumor growth control in the untreated tumor in contrast to all other groups (**Figure 7B**, left panel). As we sought to investigate the immunological modulation due to the treatments, tumors were harvested for downstream flow cytometric analysis. Interestingly, PeptiCRAd1 showed higher CD8+/CD4+ T-cell ratio (**Figure 8A**) within the TME of the treated tumor (right side) well in line with an increased CD8+ T-cell infiltration (**Figure 8B**) in both treated (right side) and untreated (left side) cancer lesions. Moreover, the improved tumor growth control achieved in the PeptiCRAd1 group correlated with the upregulation of the migratory marker CXCR4 in the CD8+ T-cell population in both treated and untreated tumors (**Figure 8C**) and upregulation of effector marker CXCR3 in the CD8+ T-cell population in the treated lesions (**Figure 8D**). Exhaustion

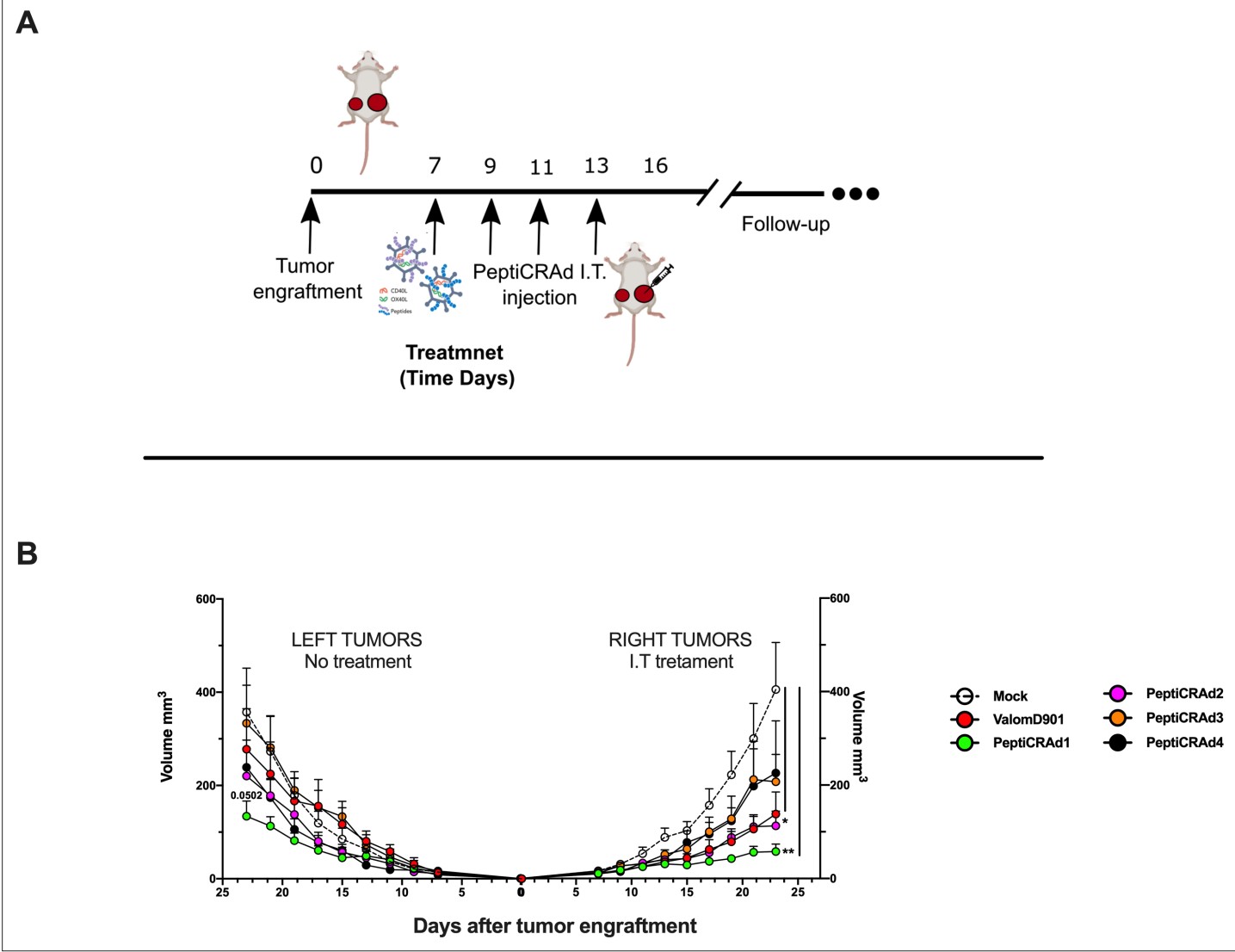

**Figure 7.** PeptiCRAd improved the tumor growth control in both injected and not injected lesions. (A) A schematic representation of the animal experiment setting is depicted. Immunocompetent Balb/c mice were subcutaneously injected with the syngeneic tumor model CT26 in the left (0.6 × 10⁶ cells) and right flanks (1 × 10⁶). PeptiCRAd was intratumorally administrated four times, 2 days apart. (B) The CT26 tumor growth was followed until the end of the experiment, and the tumor size is presented as the mean ± SEM. Statistically significant difference was assessed with two-way ANOVA (*p<0.05; ***p<0.001; ****p<0.0001; ns, nonsignificant).

The online version of this article includes the following figure supplement(s) for figure 7:

**Figure supplement 1.** Single-tumor growth for single mouse for each treatment group is depicted.

markers PD1 and TIM3 were also analyzed. The expression of PD1 in the CD8+ T-cell population showed a tendency to be upregulated in both treated and untreated cancer lesions (*Figure 8—figure supplement 1A*), suggesting the presence of antigen-experienced T-cell response. On the other hand, exhausted CD8+ T cells phenotypically defined as PD1+ and TIM3+ were downregulated in the untreated lesions; the same tendency was also seen in the treated tumors (*Figure 8—figure supplement 1B*). We further investigated the CD4+ T-cell compartment. Our oncolytic cancer vaccine treatment induced a modest downregulation of the CD4+ T cells in both treated and untreated tumors (*Figure 8—figure supplement 1C*) in line with the increase of CD8+ T cells as mentioned before. The CD4+ population showed upregulation of CXCR4 in the treated tumors in PeptiCRAd1, Pepti-CRAd2, PeptiCRAd3 compared to the VALO-mD901-treated tumors; however, no differences were observed when compared to the mock group. Even though the effector marker CXCR3 was downregulated in the untreated and treated tumors, PeptiCRAd1 showed the tendency in upregulating

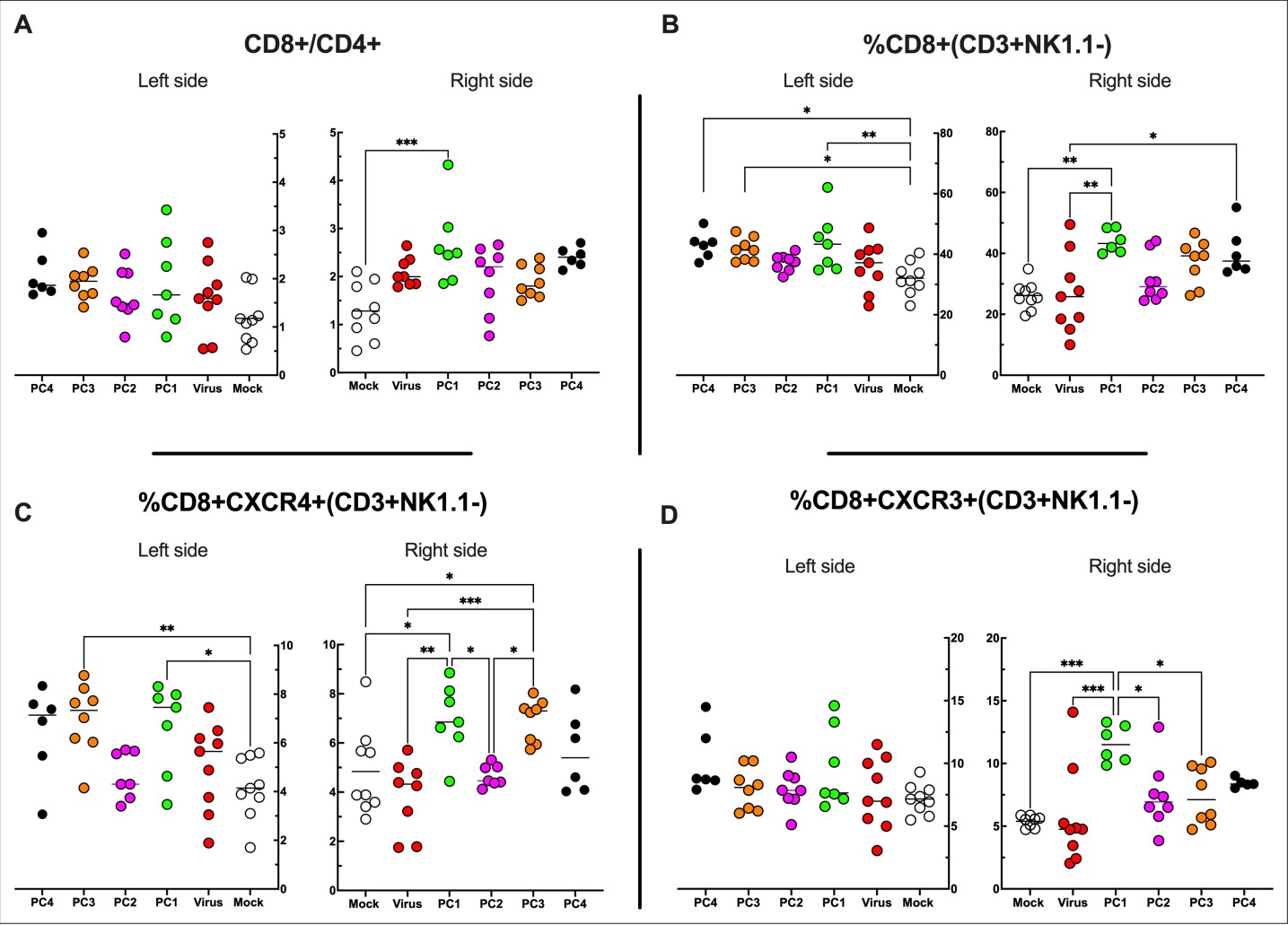

**Figure 8.** Flow cytometry analysis of tumor-Infiltrating lymphocytes (TILs). (**A–D**) The treated (right side) and untreated tumors (left side) were harvested at the end of the experiment and analyzed for the CD8+/CD4+ ratio (**A**) and the frequency of CD8+ (**B**), CD8+ CXCR4+ (**C**), and CD8+ CXCR3+ (**D**) in the tumor microenvironment (TME). All the data are plotted as dot plot for each mouse and each treatment group. The significance was assessed by one-way ANOVA and Tukey's correction (*p<0.05; ***p<0.001; ****p<0.0001; ns, nonsignificant).

The online version of this article includes the following figure supplement(s) for figure 8:

**Figure supplement 1.** Immunophenotype of tumors treated and untreated.

**Figure supplement 2.** Immunophenotyping of T-cell memory population in spleens harvested from mice upon treatments.

CXCR3 in the untreated lesion (*Figure 8—figure supplement 1C*). No statistical differences were observed in regard to the antigen experienced or exhausted phenotypes compared to the control groups (*Figure 8—figure supplement 1C*). According to previous studies, the effectiveness of the immune therapy is associated with the induction of T central memory (Tcm) or T stem cell memory (Tscm)-like CD8+ T cells in the peripheral lymphoid tissues. The Tscm population has been reported as novel memory T-cell subgroup (CD44low CD62L+) characterized by high expression amount of stem cell antigen 1 (SCA-1) and capable of self-renew (*Gattinoni et al., 2017*; *Wang et al., 2021*; *Zhang et al., 2005*). This latter characteristic makes the Tscm T CD8+ cells a desirable population to develop upon vaccination (*Gattinoni et al., 2017*). In line with that, we investigated the expression of SCA-1 in CD44low CD62L+ CD8+ T cells in spleens harvested from mock-, VALO-mD901-, and PeptiCRAd1-treated groups. According to the gating strategy reported in *Figure 8—figure supplement 1A*, we observed that SCA-1 showed a tendency to be upregulated in spleens from mice treated with Pepti-CRAd1 (*Figure 8—figure supplement 2B*), suggesting the possible generation of a long-term CD8+ T-cell population in mice upon oncolytic vaccine treatment. No differences were instead observed

in CD44+ CD62L+ (Tcm) (*Figure 8—figure supplement 2C*) or CD44+ CD62L- (T effector memory) (*Figure 8—figure supplement 2D*) compartments.

Altogether, the data showed that PeptiCRAd 1 induced remodulation of the immune cell infiltration within the TME, in particular influencing the CD8+ T-cell population.

In conclusion, the pipeline reported herein could considerably facilitate the identification, prioritization, and selection of suitable peptide candidates for cancer vaccine. Moreover, we also proposed an easy and fast adenovirus-based platform for the generation of personalized oncolytic vaccines to be combined with the selected peptides for cancer immunotherapeutic treatments. We envision that our pipeline could be applied to human clinical approaches, drastically reducing the time related to both tumor peptide selection and oncolytic vaccine generation, paving the way to precision cancer immunotherapy treatments.

## Discussion

Cytotoxic antitumor CD8+ T cells (CTLs) recognize peptides typically of 8–10 amino acids within the MHC-I complex expressed on the cellular surface, and therefore, the knowledge of these peptides is the key to design T-cell-based therapeutic cancer vaccines; indeed, their efficacy relies mostly on the choice of the antigenic peptides (*Hollingsworth and Jansen, 2019*). These peptides should be highly immunogenic, expressed exclusively on the cancer cells to avoid on-target off-tumor toxicity and tailored to the patient's specific tumor ligandome landscape. However, only a fewer if any of the tumor antigens meet those characteristics, making it very difficult to generate peptide-based vaccination technologies. Thus, the isolation and identification of MHC-I peptides and the subsequent selection criteria are of utmost importance in creating those vaccines. To fulfill these needs, we conceived a pipeline that comprises all the steps considered essential for an optimal development of a therapeutic cancer vaccine.

We decided to identify and isolate peptides directly from the MHC-I complexes, exploiting state-of-the-art immunoprecipitation and mass spectrometric methodologies as the direct elution and analysis of MHC-I-restricted peptides is so far the most reliable and used approach in studies of ligandome landscape (*Freudenmann et al., 2018*), identifying naturally processed and presented tumor epitopes that could generate clinically relevant antitumor responses. Even though computational algorithms can take into account the entire MHC complex presentation machinery (e.g., proteasomal cleavage, transporter-associated antigen processing [TAP] transport, binding motif) to predict relevant T-cell epitopes, the lack of validated and homogenous datasets makes the process difficult and less reproducible (*Feola et al., 2020*; *Soria-Guerra et al., 2015*). These considerations prompted us to adopt direct MHC-I immunoaffinity purification as a first step of our pipeline.

Moreover, to develop and further validate our proof-of-concept pipeline, the choice of the tumor model needed to meet specific requirements. First, we wanted a tumor model that expresses sufficient levels of MHC-I complexes, granting a fruitful recovery of peptides from the cellular surface. Indeed, the overall idea was to obtain a conspicuous list of peptides in order to later on challenge our prioritizing and selection criteria. Secondly, to test the selected candidates for their antitumoral efficacy profile, the preclinical model should have beneficial immunogenic features, in particular T-cell infiltration into the TME, allowing a better study of the immune modulation upon treatment administration. Based on that, the colon tumor model CT26 was selected as it showed high expression level of MHC-I complexes as demonstrated in our flow cytometry analysis and for being a widely used and characterized tumor model for developing and testing immunotherapeutic concepts in vivo (*Taylor et al., 2019*; *Castle et al., 2014*). As expected, the immunoaffinity purification generated a long list of peptides, containing more than 8000. Before moving forward in our pipeline, we carefully analyzed the quality of the produced dataset to ensure the solidity of our list and examine it for the presence of contaminants. The analysis demonstrated that the eluted peptides resembled a typical ligandome profile and therefore they could be considered as true MHC-I ligands. The strength and reliability of the ligandome dataset are critically important for the following steps as they influence the subsequent results.

Of note, besides the identification of MHC-I peptides, the main issue is dealing with the prioritization of the peptides among thousands of possible candidates. In this context, we followed two parallel directions. First, we adopted a more conservative approach that consisted of analyzing the RNAseq expression level of the respective source proteins. In particular, based on the definition of

TAA as an antigen overexpressed in malignant cells compared to healthy tissue, we considered the colon Balb/c as the reference normal tissue since CT26 is an undifferentiated colon carcinoma induced by the carcinogen *N*-nitroso-*N*-methylurethane (NMU) (*Castle et al., 2014*). In this sense, the selected peptides used as therapeutic cancer vaccine should evoke specific antitumor CTLs able to recognize and eradicate tumor cells, avoiding damages to normal colon tissue. Moreover, syngeneic mTECs express most of the known genes, and they are the site of T-cell selection to induce central tolerance to MHC peptides coded by their vast transcriptome. We assumed that the breaking of tolerance could most likely happen if the source proteins of the selected peptides from our dataset were over-represented in the CT26 cell line compared to the mTEC. To ensure a more accurate selection of the candidates, we focused the choice on peptides that meet both criteria of (1) source protein overex-pressed compared to normal colon and mTEC and (2) of being true MHC-I strong ligands. The second parallel approach represents the main novelty introduced in our pipeline, and it consisted of selecting peptides based on their similarity to antigen pathogen by exploiting HEX, a tool previously developed in our laboratory and successfully validated both in preclinical and clinical settings (*Jacopo, 2020*). The main idea relies on the intrinsic degeneracy of the TCR, defined as the ability of a single TCR to recognize more than one antigen, generating a phenomenon known as cross-reactivity. This property is an essential feature to broaden the breadth of the T-cell repertoire, and, for instance, it allows antiviral memory CD8+ T cells generated by prior infections to recognize unrelated viruses, as demonstrated in several studies in human and murine models (*Welsh and Selin, 2002*; *Brehm et al., 2004*). We thought that the same concept could be applied to cancer antigens that have similarities to viral antigens. We are aware that in this work we used mice naïve to viral infections, and therefore, no memory CD8+ T-cell cross-reactivity could be exploited. However, translated into a real clinical setting, our approach will have the added value of exploiting the cross-reactivity of preexisting viral CD8+ T cells to enhance the antitumor response. Applying the aforementioned in silico analysis, the number of candidates was shortened, making it feasible to further functionally characterize the list of the peptides in an ELISpot assay.

After the selection of candidates to exploit in a vaccine platform was done, we employed the peptides in our previously developed platform named PeptiCRAd, an oncolytic adenovirus coated with polyK-modified peptides (*Capasso et al., 2016*; *Feola et al., 2018*; *Ylösmäki et al., 2021*; *Tähtinen et al., 2020*). Indeed, after the FDA approval of T-VEC, a herpes virus encoding GM-CSF (*Ribas et al., 2018*) for the treatment of melanoma, the use of oncolytic viruses (OVs) has been extensively explored in cancer immunotherapy (*Kaufman et al., 2016*; *Harrington et al., 2019*; *Macedo et al., 2020*; *Kottke et al., 2021*). OVs are naturally occurring or genetically modified viruses able to infect and replicate in cancer cells; the OVs induce a systematic immune response, involving both innate and adaptive immune response. Moreover, the antigen spread following the viral burst acts as in situ cancer vaccine but is often not enough to generate a specific antitumor adaptive immune response, instead generating mainly an antiviral T-cell responses (*Ylösmäki and Cerullo, 2020*). To overcome this limitation, we decided to combine the immunogenicity of the OVs with the antitumor specificity of the peptides, generating an oncolytic cancer vaccine. Thus, to challenge our pipeline and investigate whether our selection criteria could actually be used to identify relevant candidates for cancer treat-ment, we decorated the OAd VALO-mD901 with the selected peptides to treat immunocompetent CT26 tumor-bearing mice. To understand whether our technology could actually evoke a systemic antitumor immune response, we engrafted two tumors for each mouse, both right and left flanks, and then treated only the tumor on the right flank. Of note, VALO-mD901 is encoding murine CD40L and OX40L under cytomegalovirus (CMV) promoter allowing transgene expression in murine cells. Stimulation of innate (due to CD40L) and adaptive (due to OX40L) immune cells explained the local antitumor activity in virus-injected tumors observed in our results and the lack of efficacy in the distant lesions. Contrarily, PeptiCRAd 1 (virus coated with peptide 1) treatment slowed the tumor growth of both the treated and untreated lesions, highlighting the generation of a systemic tumor-specific immune response. Additionally, in order to benchmark our pipeline, we employed as positive control peptide 7, an immunodominant peptide in CT26 cell line named AH1 (gp70$_{423–431}$); however, AH1 elicits variable antitumor response because T cells are suboptimally activated. To this end, we employed the heterolytic version named AH1-5, which differs from the AH1 peptide in a single amino acid (val to ala in position 5) that has been reported to protect mice against subsequent tumor challenge with CT26 when employed in a prophylactic setting (*Jordan et al., 2010*). Based on that and in the absence of

other positive candidate controls, we reasoned that AH1-5 could be used in our work as an internal control. Nevertheless, in our work, we used a therapeutic approach treating well-established tumors instead of a prophylactic setting as showed in *Jordan et al., 2010*, and in conclusion our data showed that AH1-5 was actually ineffective against well-established tumors.

The overall results demonstrated the feasibility of applying the described pipeline for the generation of a tailored therapeutic cancer vaccine. We have addressed all the main issues universally recognized as challenges in the field with main focus on the prioritization and selection criteria among thousands of peptide candidates. Additionally, we adapted quick 'plug-and-play' technology based on decorating an OV with the selected peptides. The nature of this technology opens the possibility of a fast generation of tailored therapeutic cancer vaccines in future clinical applications where personalized therapies represent one of the main goals for a successful treatment. From a clinical application point of view, the integration of the ligandome and transcriptome analysis could benefit from the fast selection of peptides done with the HEX software. Indeed, recent data suggest that MHC-I-restricted peptides homologous to viral peptides are strongly immunogenic and offer a reliable source of candidates for cancer vaccine design. Our approach will capitalize on preexisting cross-reactive T cells (*Zitvogel et al., 2021*; *Fluckiger et al., 2020*), facilitating the peptide selection.

## Methods
### Cell lines and reagents
Murine colon carcinoma CT26 cell line was purchased from ATCC (ATTC CRL-2639) and cultured in RPMI-1640 supplemented with 1% GlutaMAX (Gibco, Invitrogen, Carlsbad, CA), 10% heat-inactivated fetal bovine serum (Gibco), and 1% penicillin-streptomycin (10,000 U/mL) (Gibco). The cells were cultivated at 37°C, 5% $CO_2$ in a humidified atmosphere and routinely tested for mycoplasma contamination.

Poly(I:C) (HMW) VacciGrade 10 mg was obtained from Invivogen (San Diego, CA).

The following peptides were used for the pre-immunization experiment: SYHPALNAI, SYLTSASSL, YYVRILSTI, SYLPPGTSL, RYLPAPTAL, KYIPAARHL, AFHSSRTSL, NYNSVNTRM, SYSDMKRAL, FYEKNKTLV, KGPNRGVII, FYKNGRLAV, LYKESLSRL, SYRDVIQEL, KFYDSKETV, KYLNVREAV, HYLP-DLHHM, SGPNRFILI, SYIIGTSSV, RGPYVYREF, FYATIIHDL, GYMTPGLTV, SYLIGRQKI, AGASRIIGI, QGPEYIERL, and SYIHQRYIL.

All peptides were purchased from Zhejiang Ontores Biotechnologies (Zhejiang, China).

The following peptides were used through the animal study and purchased from PepScan (Lely-Stand, the Netherlands): KKKKKKSYLPPGTSL (Mavs), KKKKKKRYLPAPTAL (Fanca), KKKKKKYIPAARHL (Zw10), KKKKKKLYKESLSRL (Myh14), KKKKKKYLNVREAV (Chac1), KKKKKKKFYATIIHDL (Ndst3), SYLP-PGTSL (Mavs), RYLPAPTAL (Fanca), KYIPAARHL (Zw10), LYKESLSRL (Myh14), KYLNVREAV (Chac1), annd FYATIIHDL (Ndst3).

### Oncolytic adenovirus
In this study, the virus VALO-mD901 was used, and it was generated according to *Ylösmäki et al., 2021*. Briefly, VALO-mD901 is a conditionally replicating adenovirus serotype 5 with adenovirus 3 fiber knob modification and 24-base pair deletion of the gene E1A. The E3 region was replaced with human CMV promoter region, murine OX40L, 2A self-cleaving peptide sequence, murine CD40Lgene, and β-rabbit globin polyadenylation signal. The VP concentration was measured at 260 nm, and infections units were determined by immunocytochemistry by staining the hexon protein on infected A549 cells.

### IFN-γ ELISpot
IFN-γ ELISpot assays were performed using a commercially available mouse ELISpot reagent set (ImmunoSpot, Bonn, Germany), and 20 ng/μL of each peptide was tested in in vitro stimulations of $3 \times 10^5$ splenocytes for each well at 37 °C for 72 hr. Spots were counted using an ELISpot reader system (ImmunoSpot).

### PeptiCRAd complex formation
The PeptiCRAd complex was prepared by mixing the oncolytic adenovirus VALO-mD901 and each peptide with a polyK tail. We mixed polyK-extended epitopes with VALO-mD901 for 15 min at room

temperature prior to treatments with the PeptiCRAd complexes. More details about the stability and formation of the complex can be found in our previous study (*Capasso et al., 2016*).

## Animal experiment

All animal experiments were reviewed and approved by the Experimental Animal Committee of the University of Helsinki and the Provincial Government of Southern Finland (license number ESAVI/11895/2019). 4- to 6-week-old female Balb/cOlaHsd mice were obtained from Envigo (Laboratory, Bar Harbor, Maine, UK).

For the pre-immunization experiment, mice (n = 3 per group) were allocated in nine different groups and each mouse was injected three times (one injection for each peptide) in three different areas (each injection contained 25 µg of peptide + 25 µg of poly (I:C)). The prime and boosting were done respectively at days 0 and 7, and the mice were sacrificed at day 14.

For the tumor-bearing mice experiment, $1 \times 10^6$ and $6 \times 10^5$ CT26 cells were injected subcutaneously into the right and left flanks, respectively. A complete randomization was done on the day of the treatment. Details about the schedule of the treatment can be found in the figure legends. Viral dose was $1 \times 10^9$ vp/tumor complexed with 20 µg of a single peptide or with 10 µg + 10 µg mixture of two peptides.

## Flow cytometry

The antibodies were TruStain FcX anti-mouse CD16/32 (BioLegend), APC-H2Kd (BioLegend), BV711-CD3 (BD Horizon), PE-CF594-CD4 (BD Horizon), FITC-NK1.1 (Invitrogen), PE-PD1 (BioLegend), APC-CXCR3 (BD Pharmigen), PE-CY7-TIM3 (BioLegend), BV510-CD8 (BD Horizon), and V450-CXCR4 (BD Horizon).

The data were acquired using BD LSRFortessa flow cytometer and analyzed using FlowJo software v9 (Ashland, OR).

## Purification and concentration of MHC-I peptides

MHC class I peptides were immunoaffinity purified from the CT26 mouse cell line using anti-mouse MHC class I (clone 34-1-2S, BioXCell, BE0180, Lebanon, USA). For sample preparation, the snap-frozen cell pellet ($1 \times 10^8$ cells for each replicate, in total six replicates) was incubated for 2 hr at 4°C in lysis buffer. The lysis buffer contained 150 mM NaCl, 50 mM Tris-HCl, pH 7.4, protease inhibitors (A32955, Thermo Scientific Pierce, Waltham, MA), and 1% Igepal (I8896, Sigma-Aldrich, St. Louis, MO). The lysates were first cleared by low-speed centrifugation for 10 min at 500 × *g*, and then the supernatant was centrifuged for 30 min at 25,000 × *g*. Next, MHC-I complexes were immunoaffinity purified loading the cleared lysate to the immunoaffinity column (AminoLink Plus Immobilization, Pierce) with covalently linked antibody according to the manufacturer's instructions. Following binding, the affinity column was washed using seven column volumes of each buffer (150 mM NaCl, 20 mM Tris-HCl; 400 mM NaCl, 20 mM Tris-HCl; 150 mM NaCl, 20 mM Tris-HCl, and 20 mM Tris-HCl, pH 8.0) and bound complexes were eluted in 0.1 N acetic acid.

Eluted HLA peptides and the subunits of the HLA complexes were desalted using SepPac-C18 cartridges (Waters) according to the protocol previously described by *Bassani-Sternberg, 2018*. Briefly, the cartridge was prewashed with 80% acetonitrile in 0.1% trifluoroacetic acid (TFA) and then with 0.1% TFA. The peptides were purified from the MHC-I complex by elution with 30% acetonitrile in 0.1% TFA. Finally, the samples were dried using vacuum centrifugation (Eppendorf).

## Algorithms used for prediction of peptide ligands

Affinity to the H2Kd/H2Dd alleles was predicted for all eluted peptides identified in the CT26 cell line using NetMHC4.0 (*Andreatta and Nielsen, 2016*, *Nielsen et al., 2003*). The threshold for binding was set to rank 2% to include only the binding partners.

## Gibbs clustering analysis

Clustering of peptides into groups based on sequence similarities was performed using the Gibbs-sCluster-2.0 tool with the default settings (*Andreatta et al., 2017*; *Andreatta et al., 2013*).

## GO enrichment analysis

ClusterProfiler Bioconductor package (v. 3.12.0) in the RStudio server environment (v. 3.6.0) (*Yu et al., 2012*) was used for the functional annotation and visualization. ClusterProfiler implements a hypergeometric test to evaluate the statistical enrichment of the input gene list over the desired functional classes.

## DESeq profile

Raw sequence data for colon tissue (source: GEO accession #GSE92563) and mTEC/CT26 (source: GEO accession: #GSE111092) were mapped to the mouse genome Mus_musculus GRCm38.95 using the online tool Chipster (*Kallio et al., 2011*).

Briefly, FastQ files were combined for each sample sequencing using the function 'Make a list of file names: paired end data.' The alignment to the reference genome and the count aligned reads per gene was done respectively with HISAT2 and HTSeq. Finally, the differential expression analysis used DESeq2, applying a cutoff for the adjusted p-value of 0.05 (Benjamini–Hochberg adjusted p-value). The 'MultiQC function' was used to assess the quality of the FastQ files.

## LC-MS analysis of MHC-I peptides

Each dry sample was dissolved in 10 µL of LCMS solvent A (0.1% formic acid) by dispensing/aspirating 20 times with the micropipette. The nanoElute LC system (Bruker, Bremen, Germany) injected and loaded 10 µL of the sample directly onto the analytical column (Aurora C18, 25 cm long, 75 µm ID, 1.6 µm bead size, Ionopticks, Melbourne, Australia) constantly kept at 50°C by a heating oven (PRSO-V2 oven, Sonation, Biberach, Germany). After washing and loading the sample at a constant pressure of 800 bar, the LC system started a 30 min gradient from 0% to 32% solvent B (acetonitrile, 0.1% formic acid), followed by an increase to 95% B in 5 min, and finally a wash of 10 min at 95% B, all at a flow rate of 300 nL/min. Online LC-MS was performed using a Tims TOF Pro mass spectrometer (Bruker) with the CaptiveSpray source, capillary voltage 1500 V, dry gas flow of 3 L/min, dry gas temperature at 180°C. MS data reduction was enabled. Mass spectra peak detection maximum intensity was set to 10. Mobilogram peak detection intensity threshold was set to 5000. Mass range was 300–1100 $m/z$, and mobility range was 0.6–1.30 V.s/cm$^2$. MS/MS was used with three PASEF (parallel accumulation – serial fragmentation) scans (300 ms each) per cycle with a target intensity of 20,000 and intensity threshold of 1000, considering charge states 0–5. Active exclusion was used with release after 0.4 min, reconsidering precursor if the current intensity is greater than fourfold the previous intensity, and a mass width of 0.015 $m/z$ and a 1/k0 width of 0.015 V.s/cm$^2$. Isolation width was defined as 2.00 $m/z$ for mass 700 $m/z$ and 3.00 $m/z$ for mass 800 $m/z$. Collision energy was set as 10.62 eV for 1/k0 0.60 V.s/cm$^2$ and 51.46 eV for 1/k0 1.30 V.s/cm$^2$. Precursor ions were selected using 1 MS repetition and a cycle overlap of 1 with the default intensities/repetitions schedule.

## Proteomics database search

All MS/MS spectra were searched by PEAKS Studio X+ (v10.5 build 16 October 2019) using a target-decoy strategy. The database used was the Swissprot Mouse protein database (including isoforms, 25,284 entries, downloaded from uniprot.org on 27 November 2019).

A precursor mass tolerance of 20 ppm and a product mass tolerance of 0.02 Da for CID-ITMS2 were used. Enzyme was none, digest mode unspecific, and oxidation of methionine was used as variable modification, with max three oxidations per peptide. An FDR cutoff of 1% was employed at the peptide level. The mass spectrometry proteomics data have been deposited to the ProteomeXchange Consortium via the PRIDE partner repository with the dataset identifier PXD026463.

## Surface plasmon resonance

Measurements were performed using a multiparametric SPR Navi 220A instrument (Bionavis Ltd, Tampere, Finland). Phosphate-buffered saline (PBS) (pH 7.4) was used as a running buffer, a constant flow rate of 20 µL/min was used throughout the experiments, and temperature was set to +20°C. Laser light with a wavelength of 670 nm was used for surface plasmon excitation and analysis. APTES-coated Au-SiO$_2$ sensor slides were used to immobilize VALO-mD901 viruses on the sensors for evaluating peptide affinity and assessing the number of peptides per VALO-mD901 virus. The APTES-coated Au-SiO$_2$ was prepared by first activating its surface by 5 min of oxygen plasma treatment

followed by incubating the sensor in 50 mM APTES in isopropanol for 4 hr, thus rendering the SPR sensor highly positively charged. The sensor was then washed and placed into the SPR device. The VALO-mD901 viruses were immobilized in situ on the sensor surface by injecting approximately $4.96 \times 10^{11}$ vp/mL in PBS (pH 7.4) for 10 min, followed by a 10 min wash with PBS. For testing the interaction between various peptides and the immobilized VALO-mD901 viruses, 100 µM of the tested peptides were injected onto the viruses.

The SPR responses measured during virus immobilization as well as peptide interactions were used to estimate how many peptides were adsorbed per virus. This estimation is based on geometrical calculations including the SPR detection area ($A_S = \pi r^2$, where $r = 0.5$ mm), diameter of the virus ($d = 100$ nm), footprint area one virus covers on the SPR sensor ($A_V = \pi r^2$, where $r = 50$ nm), SPR signal response for a sensor fully covered with viruses ($\Delta° = 1.4°$), percent coverage of viruses in the detection area ($C(\%) = $ (measured SPR response)/(SPR response for full layer of viruses, i.e., 1.4°)), area covered by viruses in the detection area ($A_{V,cov} = A_S \times C(\%)$), number of viruses in detection area ($N_V = A_{V,cov}/A_V$), mass/area of peptides determined from the corresponding SPR response (m/A = measured SPR response $\times$ 660 ng/cm$^2$), mass of peptides in the detection area ($m_P = m/A \times A_S$), and the number of peptides in the detection area ($N_P = [(m_P/M_P) \times N_A]$, where $M_P$ is the molecular weight of the peptide and $N_A$ is the Avogadro constant).

## Statistical analysis

Statistical analysis was performed using GraphPad Prism 9.0 software (GraphPad Software Inc). Details about the statistical tests for each experiment can be found in the corresponding figure legends.

## Acknowledgements

We thank all the participants for their support and advice. Moreover, flow cytometry analysis was performed at the HiLife Flow Cytometry Unit, University of Helsinki. This work has been supported by the European Research Council under the European Union's Horizon 2020 Framework programme (H2020)/ERC-CoG-2015 Grant Agreement No. 681219, the Helsinki Institute of Life Science (HiLIFE), the Jane and Aatos Erkko Foundation (decision 19072019), the Cancer Society of Finland (Syöpäjärjestöt), ERC (POC), Business Finland. CB received funding from Associazione Italiana per la Ricerca sul Cancro (AIRC-IG 18458 and AIRC 5 per Mille 22737), the Italian Ministry of Education, University and Research (PRIN 2017WC8499), and the Italian Ministry of Health and Alliance Against Cancer (Ricerca Corrente CAR T project: RCR-2019-23669115), EU (T2Evolve); ER received funding from the Italian Ministry of Health (GR-2016-02364847).

## Additional information

### Competing interests

Sari Pesonen: is an employee and a shareholder at VALO Therapeutics. The other authors declare that no competing interests exist.

### Funding

| Funder | Grant reference number | Author |
|---|---|---|
| Ministry of Health | | Eliana Ruggiero |
| European Research Council | | Vincenzo Cerullo |
| Associazione Italiana per la Ricerca sul Cancro | | Chiara Bonini |
| Horizon 2020 | | Vincenzo Cerullo |
| Business Finland | | Vincenzo Cerullo |

The funders had no role in study design, data collection and interpretation, or the decision to submit the work for publication.

## Author contributions

Sara Feola, Conceptualization, Data curation, Formal analysis, Investigation, Methodology, Project administration, Resources, Software, Supervision, Validation, Visualization, Writing - original draft, Writing – review and editing; Jacopo Chiaro, Data curation, Investigation, Writing – review and editing; Beatriz Martins, Conceptualization, Data curation, Writing – review and editing; Salvatore Russo, Investigation, Methodology, Writing – review and editing; Manlio Fusciello, Firas Hamdan, Michaela Feodoroff, Investigation, Methodology; Erkko Ylösmäki, Conceptualization, Data curation, Formal analysis, Investigation, Methodology; Chiara Bonini, Eliana Ruggiero, Methodology, Validation; Gabriella Antignani, Data curation, Methodology; Tapani Viitala, Data curation, Visualization, Writing – review and editing; Sari Pesonen, Data curation, Visualization, Writing - original draft, Writing – review and editing; Mikaela Grönholm, Data curation, Writing - original draft, Writing – review and editing; Rui MM Branca, Data curation, Methodology, Writing – review and editing; Janne Lehtiö, Methodology, Writing – review and editing; Vincenzo Cerullo, Conceptualization, Funding acquisition, Project administration, Writing - original draft, Writing – review and editing

## Author ORCIDs

Sara Feola ⓘ http://orcid.org/0000-0002-4012-4310
Michaela Feodoroff ⓘ http://orcid.org/0000-0002-6094-9838
Tapani Viitala ⓘ http://orcid.org/0000-0001-9074-9450
Janne Lehtiö ⓘ http://orcid.org/0000-0002-8100-9562
Vincenzo Cerullo ⓘ http://orcid.org/0000-0003-4901-3796

## Ethics

All animal experiments were reviewed and approved by the Experimental Animal Committee of the University of Helsinki and the Provincial Government of Southern Finland (license number ESAV-I/11895/2019).4-6 weeks old female Balb/cOlaHsd mice were obtained from Envigo (Laboratory, Bar Harbor, Maine UK).

## Decision letter and Author response

Decision letter https://doi.org/10.7554/eLife.71156.sa1
Author response https://doi.org/10.7554/eLife.71156.sa2

---

# Additional files

## Supplementary files

• Supplementary file 1. List of candidate peptides derived by differential gene expression profile (DESeq) analysis in CT26 versus (medullary thymic epithelial cell (mTEC) and CT26 versus healthy Balb/c colon). For each peptide, the Uniprot ID, gene name, and sequence are reported. Additionally, the last column indicates whether (1) or not (0) the peptide has been already described in a published ligandome dataset.

• Supplementary file 2. HEX software results. For each peptide, the Uniprot ID, amino acid sequence, and similar pathogen species with the respective viral peptides with sequence similarity are shown. The last column indicates whether (1) or not (0) the peptide has been already described in a published ligandome dataset.

• Supplementary file 3. List of peptides tested in the enzyme-linked immunospot (ELISpot) assay. For each group of mice, the peptides with the respective identification number as indicated in the ELISpot assay are reported.

• Supplementary file 4. List of peptides with the respective polyK-peptides. The candidate peptides used in PeptiCRAd technology with the respective net charge without and with the poly-lysine modification are shown.

• Supplementary file 5. List of polyK-peptides used in PeptiCRAd. The poly-lysine-modified peptides, Uniprot ID, and respective gene names for each PeptiCRAd treatment group are summarized.

• Transparent reporting form

## Data availability

The mass spectrometry proteomics data have been deposited to the ProteomeXchange Consortium via the PRIDE partner repository with the dataset identifier PXD026463.

The following dataset was generated:

| Author(s) | Year | Dataset title | Dataset URL | Database and Identifier |
|---|---|---|---|---|
| Feola S, Chiaro J, Martins B, Russo S, Fusciello M, Ylösmäki E, Bonini C, Ruggiero E, Hamdan F, Feodoroff M, Antignani G, Viitala T, Pesonen S, Grönholm M, Branca RMM, Lehtiö J, Cerullo V | 2021 | A novel immunopeptidomic-based pipeline for the generation of personalized oncolytic cancer vaccines | https://www.ebi.ac.uk/pride/archive/projects/PXD026463 | PRIDE, PXD026463 |

The following previously published datasets were used:

| Author(s) | Year | Dataset title | Dataset URL | Database and Identifier |
|---|---|---|---|---|
| Laumont CM, Vincent K, Hesnard L, Audemard E, Bonneil E, Laverdure JP | 2018 | Noncoding regions are the main source of targetable tumor-specific antigens | https://www.ncbi.nlm.nih.gov/geo/query/acc.cgi?acc=GSE111092 | NCBI Gene Expression Omnibus, GSE111092 |
| Proquin H, Jetten MJ, Jonkhout MCM, Garduño-Balderas LG | 2018 | Transcriptomics effects of ingestion of food additive titanium dioxide (E171) in the colon of BALB/c mice | https://www.ncbi.nlm.nih.gov/geo/query/acc.cgi?acc=GSE92563 | NCBI Gene Expression Omnibus, GSE92563 |

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
