## [Editor Report]

The reviewers find that the findings presented in this study are interesting and have an immediate impact on immune oncology.

---

## [Decision Letter]

**Decision letter after peer review:**

Thank you for submitting your article "A novel immunopeptidomic-based pipeline for the generation of personalized oncolytic cancer vaccines" for consideration by *eLife*. Your article has been reviewed by 3 peer reviewers, one of whom is a member of our Board of Reviewing Editors, and the evaluation has been overseen by Carla Rothlin as the Senior Editor. The following individual involved in the review of your submission has agreed to reveal their identity: Etienne Caron (Reviewer #2).

Essential revisions:

1) Improving the data analyses with proper controls.

2) Testing in additional setting, such as human samples or MCA sarcomas for fulfilling the claim on personalised format and validating the power of this platform.

3) A clear therapeutic advantage of exploiting molecular mimicry and tumor pathogen cross-reactive T cells over conventional TSA.

*Reviewer #2 (Recommendations for the authors):*

In this study, Feola et al., describe an immunopeptidomic-based pipeline to identify new tumor antigens, which are used in a vaccine platform to treat mice bearing established colon tumors. This study is technical rather than conceptual and aims to exploit viral molecular mimicry and tumor pathogen cross-reactive T-cells in cancer vaccine development. Please find below my concerns.

1) The most important aspect of this technical paper is the assessment of the therapeutic potential of the cancer vaccines to control tumor growth in vivo (Figure 7). However, the results are confusing and difficult to interpret, raising doubts about the overall approach.

i) In fact, the PeptiCRAd4 (ValomD901+gp70423-431 peptide) is the positive control since gp70423-431 is known to be immunodominant according to the authors. It is unclear how well this positive control was supposed to perform, and based on the results, tumor growth is hardly controlled using this immunodominant peptide.

ii) The uncoated adenovirus (ValomD901; no peptide), which I assumed was used as a negative control, outperformed PeptiCRAd3 and PeptiCRAd4 in terms of controlling tumor growth. This is confusing and suggest that the efficiency of such cancer vaccines can be MHCI-peptide independent.

iii) Considering point (i) and (ii), would other controls be more appropriate? How should the readers interpret such results? It seems to me that the final outcome is unpredictable and difficult to interpret. In other words, the current work gives the impression that if we are lucking, we may get relatively good tumor growth control, but it might also get worse than a placebo/negative control. The immediate impact of the current vaccine platform is therefore limited.

2) The viral molecular mimicry is very interesting but difficult to fully appreciate in this manuscript since very little information is provided about the HEX software and viral molecular mimicry in general. For instance, what is the state-of-the-art in exploiting molecular mimicry and tumor pathogen cross-reactive T-cells in cancer vaccine development? This should be introduced. Also, I have the impression that the opportunity that is missed in this paper to increase its impact would be the comparison of the newly identified HEX software-derived peptides versus previously reported TSA from the same tumor cell line. The immunopeptidome of CT26 cells has been characterized in Laumont et al., (2018) (PMID: 30518613). In Laumont et al., mutated TSA (mTSA), aberrantly expressed TSA (aeTSA) and conventional tumor-associated antigen (TAA) have been identified. Assessing the therapeutic potential of those TSA/TAA versus tumor antigens similar to pathogen antigens in the context of the vaccine platform described by the authors would be quite informative and may have the potential to further put forth the relevance of exploiting viral molecular mimicry in cancer immunotherapy.

In summary, I think this is a nice technical paper (with a lot of work!) but I am not convinced about the immediate impact of the current pipeline. The pipeline would benefit many other researchers in the field of cancer immunotherapy if the authors and/or other groups would show (1) a clear therapeutic advantage of exploiting molecular mimicry and tumor pathogen cross-reactive T cells over conventional TSA, and (2) a better mechanistic understanding of pathogen-cross reactive T cells in cancer.

*Reviewer #3 (Recommendations for the authors):*

The authors should test this system against spontaneous cancers, such as carcinogen-induced tumor models (i.e., MCA). By taking a resected tumor from one of the multiple induced lesions to run it through the pipeline, followed by efficacy testing of the generated vaccine, would have been a great approach to validate their pipeline and to achieve their goal for the study.

Alternatively, utilization of transplantable tumors with aggressive metastatic characteristics, such as the 4T1 breast cancer model, would have been an ideal model as well (PMID: 27584043).

This model allows resection of the primary tumor from the mammary gland, which extends the survival of the challenged mice by a month, during which the resected tumors can be used through the pipeline to develop the "personalized" oncolytic vaccines for testing their efficacy in each mouse.

By using such a system, the authors can use the cell lines as they did with CT26 while being able to use in vivo grown primary tumors as the starting material to truly mimic the clinical setting. As metastasized tumors may undergo additional immunoediting differently in each mouse, it may result in diminished expression of the peptide that was isolated from in vitro grown cancer cell line and may render generated vaccines ineffective.

Furthermore, while the authors initially demonstrated the ability of the vaccines to induce IFNγ responses in vivo by ELISPOT, they do not assess the changes in the immune responses in the treated mice beyond the surface markers of CD4 and CD8 T cells. It lacks the analysis of how the vaccines may or may not have altered the effector functions of the tumor-infiltrating immune cells. Previous studies by others have demonstrated that the induction of Tcm- or Tscm-like CD8 T cells in the peripheral lymphoid tissues is associated with effective immunotherapy responses. In order to demonstrate the long-term protection of the mice, such immunophenotyping and functional analysis may be required.

---

## [Author Response]

Reviewer #2 (Recommendations for the authors):In this study, Feola et al., describe an immunopeptidomic-based pipeline to identify new tumor antigens, which are used in a vaccine platform to treat mice bearing established colon tumors. This study is technical rather than conceptual and aims to exploit viral molecular mimicry and tumor pathogen cross-reactive T-cells in cancer vaccine development. Please find below my concerns.1) The most important aspect of this technical paper is the assessment of the therapeutic potential of the cancer vaccines to control tumor growth in vivo (Figure 7). However, the results are confusing and difficult to interpret, raising doubts about the overall approach.i) In fact, the PeptiCRAd4 (ValomD901+gp70423-431 peptide) is the positive control since gp70423-431 is known to be immunodominant according to the authors. It is unclear how well this positive control was supposed to perform, and based on the results, tumor growth is hardly controlled using this immunodominant peptide.

We thank the reviewer for giving us the possibility to discuss further and clarify the use of PeptiCRAd4 (ValomD901+gp70423-431 peptide) as “positive” control in the animal

experiment.

The epitope AH1 used as positive control is a known immunodominant antigen of CT26 derived from gp70, an endogenous viral envelope glycoprotein of murine leukemia virus (1, 2). Based on that, we decided to use it as “positive” control. In agreement with previous studies (1, 3, 4) the peptide AH1 elicits variable anti-tumor response as T cell are sub-optimally activated by AH1 when used in a vaccination setting. However, in Jordan et al., (5) a mutated version of the epitope which differs from the AH1 peptide in a single amino acid (val to ala in position 5, named AH1-5), was used to vaccinate mice. AH1-5 protected against subsequent tumor challenge with CT26. Accordingly, we reasoned that AH1-5 could be used in our work as positive internal control to benchmark our pipeline, in absence of other suitable positive controls. Nevertheless, in our work we were proposing a therapeutic approach to treat well-established tumors; instead Jordan et al., (5) were using the AH1-5 peptide in a prophylactic setting. Additionally, in Jordan et al., (5) , the challenge experiment employed 5x10^4^ cells subcutaneously injected, whereas in our pipeline we subcutaneously injected 0.6x10^6^ cells on the left and 1x10^6^ cells on the right flank for each mouse. In conclusion, our data showed that AH1-5 was ineffective against established tumors, and we hope that our results could guide other researchers in the future in the selection of immunodominant antigen for experiments performed with CT26 tumor model.

To clarify the results, we have added the following sentence in the Discussion section

“Additionally, in order to benchmark our pipeline, we employed as positive control peptide 7, an immunodominant peptide in CT26 cell line named AH1 (gp70423–431); however, AH1 elicits variable anti-tumor response because T cell are sub-optimally activated. To this end, we employed the heterolytic version named AH1-5 which differs from the AH1 peptide in a single amino acid acid (val to ala in position 5) and that has been reported to protect mice against subsequent tumor challenge with CT26 when employed in a prophylactic setting (31). Based on that and in absence of other positive candidate controls, we reasoned that AH1-5 could be used in our work as an internal control. Nevertheless, in our work we are using a therapeutic approach treating well-established tumors instead of a prophylactic setting as showed in Jordan et al., (31)

and in conclusion our data showed that AH1-5 was actually ineffective against well-established tumors.”

We hope that this aspect is now better explained in our manuscript.

ii) The uncoated adenovirus (ValomD901; no peptide), which I assumed was used as a negative control, outperformed PeptiCRAd3 and PeptiCRAd4 in terms of controlling tumor growth. This is confusing and suggest that the efficiency of such cancer vaccines can be MHCI-peptide independent.

We agree with the reviewer´s concern regarding the use of ValomD901 as negative control that outperformed PeptiCRAd3 and PeptiCRAd4 in terms of controlling tumor growth. To better clarify this point, we added the following sentence: “Of note, VALO-mD901 is encoding murine CD40L and OX40L under CMV promoter allowing transgene expression in murine cells. Stimulation of innate (due to CD40L) and adaptive (due to OX40L) immune cells explained the local anti-tumor activity in virus-injected tumors observed in our results and the lack of efficacy in the distant lesions” (Line 400-404).

iii) Considering point (i) and (ii), would other controls be more appropriate? How should the readers interpret such results? It seems to me that the final outcome is unpredictable and difficult to interpret. In other words, the current work gives the impression that if we are lucking, we may get relatively good tumor growth control, but it might also get worse than a placebo/negative control. The immediate impact of the current vaccine platform is therefore limited.

We share the reviewer´s concern regarding the outcome of the in vivo experiment. In this manuscript, our main goal is providing a pipeline for the identification of candidate peptides and subsequently generation of cancer oncolytic vaccines. The overall idea is translating the described pipeline in human application for precise medicine treatments. According to reviewers´ suggestion, we looked for a more appropriate control and we selected two peptides previously identified in a ligandome done with CT26 as reported in Laumont et al., (6). We have repeated the animal experiment to compare the antitumor activity of PeptiCRAd1(PC1 group in the main manuscript, here indicated as PeptiCRAd) to the ValomD901 coated with the Laumont peptide (Laumont). The results confirmed the tumor growth control in both injected and not injected lesions upon PetiCRAd1 treatment and the extension of the anti-tumor response is comparable to Laumont´s group activity. See Author response image 1.

**Author response image 1. sa2fig1:** The CT26 tumor growth was followed until the end of the experiment and the tumor size is presented as the mean ± SEM and statistically difference was assessed with two-way ANOVA; (*, P < 0.05; ***, P < 0.001; ****, P < 0.0001; ns, nonsignificant).

2) The viral molecular mimicry is very interesting but difficult to fully appreciate in this manuscript since very little information is provided about the HEX software and viral molecular mimicry in general. For instance, what is the state-of-the-art in exploiting molecular mimicry and tumor pathogen cross-reactive T-cells in cancer vaccine development? This should be introduced. Also, I have the impression that the opportunity that is missed in this paper to increase its impact would be the comparison of the newly identified HEX software-derived peptides versus previously reported TSA from the same tumor cell line. The immunopeptidome of CT26 cells has been characterized in Laumont et al., (2018) (PMID: 30518613). In Laumont et al., mutated TSA (mTSA), aberrantly expressed TSA (aeTSA) and conventional tumor-associated antigen (TAA) have been identified. Assessing the therapeutic potential of those TSA/TAA versus tumor antigens similar to pathogen antigens in the context of the vaccine platform described by the authors would be quite informative and may have the potential to further put forth the relevance of exploiting viral molecular mimicry in cancer immunotherapy.

We thank the reviewer for his/her suggestion and we agree that the viral molecular mimicry is an important element to exploit for cancer immunotherapeutic approaches as also previously shown in Chiaro et al., (7) and we are happy to discuss it further. The integration of T-cell cross reactivity in cancer immunotherapy has been slow and primarily focused on avoiding off-target effects. As results, peptide selection criteria avoided similarity to self (8) rather than selecting tumor peptides based on similarity to viral ones to recall crossreactive viral T cells at the tumor site. Nevertheless, emerging data highlighted that MHC-I associated peptides homologous to viral ones are strongly immunogenic; however, to the best of our knowledge, very few works have reported that pathogen specific CD4+ and CD8+ T cells attack malignant lesions. For instance, in Fluckiger et al., the authors showed that microbiota specific T cells cross-react with tumor associated antigens across different murine tumor types (9). Furthermore, in renal and lung cancer patients, microbiota cross-reactive T cells correlated with better PD1 blockade therapy response and in melanoma patients’ microbial peptides crossreactive to tumor antigen were also reported (9). Moreover, an interesting paper focuses on analysing MHC-I peptides derived from endogenous retroelements (ERE) in 16 Blymphoblastoid cell lines; here the authors confirmed the “viral” origin of EREs, demonstrating the similarity between ERE and viral peptides. As consequence of that, the ERE are more immunogenic compared to canonical MHC-I peptide, representing attractive targets for developing cancer vaccines (10). Overall, tumor antigens homologous to pathogen peptides offer an interesting source of candidates for cancer vaccine design, that could benefit future cancer immunotherapeutic approaches.

Additionally, we thank the reviewer for giving us the opportunity of assessing the

therapeutic potential of the TSA/TAA described in Laumont at al. (6) versus tumor antigens similar to pathogen antigens in the context of our vaccine platform. To this end, we repeated the animal experiment, comparing the anti-tumor activity of the best group reported in our manuscript (PC1 group in the main manuscript, here indicated as PeptiCRAd) with a PeptiCRAd version containing Laumont peptides (Laumont). The results confirmed tumor growth control in both injected and not injected lesions upon PeptiCRAd treatment and the extension of the antitumor response was comparable to Laumont´s group activity (Author response image 1). Thus, TSA/TAA described in Laumont were able to induce anti-tumor immune response at the same extent of our candidate peptides similar to viral pathogens. However, we are clearly limited by the animal model in exploring viral pre-existing T cells as the murine immune system is naive to pathogens, lacking memory pathogens T cell that could be re-engaged as an anti-tumor response. In contrast, in human setting, our approach will have the added value of exploiting the T cell memory repertoire to enhance anti-cancer immune response.

In summary, I think this is a nice technical paper (with a lot of work!) but I am not convinced about the immediate impact of the current pipeline. The pipeline would benefit many other researchers in the field of cancer immunotherapy if the authors and/or other groups would show (1) a clear therapeutic advantage of exploiting molecular mimicry and tumor pathogen cross-reactive T cells over conventional TSA, and (2) a better mechanistic understanding of pathogen-cross reactive T cells in cancer.Reviewer #3 (Recommendations for the authors):The authors should test this system against spontaneous cancers, such as carcinogen-induced tumor models (i.e., MCA). By taking a resected tumor from one of the multiple induced lesions to run it through the pipeline, followed by efficacy testing of the generated vaccine, would have been a great approach to validate their pipeline and to achieve their goal for the study.Alternatively, utilization of transplantable tumors with aggressive metastatic characteristics, such as the 4T1 breast cancer model, would have been an ideal model as well (PMID: 27584043).This model allows resection of the primary tumor from the mammary gland, which extends the survival of the challenged mice by a month, during which the resected tumors can be used through the pipeline to develop the "personalized" oncolytic vaccines for testing their efficacy in each mouse.By using such a system, the authors can use the cell lines as they did with CT26 while being able to use in vivo grown primary tumors as the starting material to truly mimic the clinical setting. As metastasized tumors may undergo additional immunoediting differently in each mouse, it may result in diminished expression of the peptide that was isolated from in vitro grown cancer cell line and may render generated vaccines ineffective.

We agree with the reviewer´s comments regarding the need of validating our pipeline for a future personalized medicine application. Thus, we are glad to discuss further the idea of using either a carcinogen-induced tumor models (such as MCA) or a transplantable tumor with aggressive characteristics (the reviewer is wisely suggesting the murine triple negative breast cancer model 4T1). We selected this latter as we have the cell line available in the lab, and we are familiar with the subcutaneous 4T1 model in mice (13). Furthermore, we checked the publication (14) suggested here by the reviewer to plan our experiment. In this work the authors developed a simple orthotopic model of breast cancer. The 4T1 cells are first injected in the mammary fat pad and between 21-30 days the tumor is established. At this time, the tumor is removed to extend the mouse life span, but the metastasis has already spread, offering an opportunity to exploit it as model of metastasis in breast cancer (14).

Even though the described model allows the researchers to examinate the metastases

burden upon different treatment, the survival of the mice is reported been around 1 month after the tumor removal (14). When we planned the experiment, we realized that the ligandome landscape investigation performed on the resected tumors would have taken at least 1 week. After that, the analysis in vitro and in vivo of the candidate peptides would have taken an additional month. Indeed, once selected in vitro the candidates, the synthesis of the peptides could have taken up to one month. Despite the time schedule is still acceptable in human scale, in the proposed murine tumor model, it is unpracticable. Looking forward to finding a murine model to validate our truly personalized cancer vaccine approach.

Furthermore, while the authors initially demonstrated the ability of the vaccines to induce IFNγ responses in vivo by ELISPOT, they do not assess the changes in the immune responses in the treated mice beyond the surface markers of CD4 and CD8 T cells. It lacks the analysis of how the vaccines may or may not have altered the effector functions of the tumor-infiltrating immune cells. Previous studies by others have demonstrated that the induction of Tcm- or Tscm-like CD8 T cells in the peripheral lymphoid tissues is associated with effective immunotherapy responses. In order to demonstrate the long-term protection of the mice, such immunophenotyping and functional analysis may be required.

We thank the reviewers for suggesting this interesting immunophenotyping and functional analysis. Following the reviewer´s suggestions, we have further investigated the T cell population in peripheral lymphoid tissue (spleens) harvested from the mice. According to the reviewer´s suggestion, we have examined the induction of T cm (T central memory) and T scm (T stem cell memory like) CD8 in spleens harvested from Mock, VALO-mD901 and PeptiCRAd1 treated groups. The T scm population has been reported as novel memory T cell subgroup (CD44low CD62L+) characterized by high amount of stem cell antigen 1 (SCA-1) and capable of self-renew (15-17). This latter characteristic makes the T scm CD8 T a desirable population to develop upon vaccination (15). In line with that, we investigated the expression of SCA-1 in CD44low CD62L+ CD8+T cells according to the gating strategy reported in Figure8—figure supplement 2A; we observed that SCA-1 showed a tendency to be upregulated in spleens from mice treated with PeptiCRAd1(Figure8—figure supplement 2B), suggesting the possible generation of a long-term CD8 T cell population in mice upon oncolytic vaccine treatment. No differences were instead observed in CD44+ CD62L+ (T central memory) (Figure8—figure supplement 2 C) or CD44+ CD62L- (T effector memory) ( Figure8—figure supplement 2D) compartments.

This is an interesting feature that we should investigate in a long-term experiment. In

the animal setting here reported, we were limited in our time window due to ethical human endpoint for the mice bearing tumors. However, given the interesting results, the data were added in the main manuscript as Supplementary Figure ("Figure8—figure supplement 2").

The following sentence has been added in the Discussion section as well:

“According to previous studies, the effectiveness of the immune therapy is associated with the induction of Tcm (T central memory) or Tscm (T stem cell memory)-like CD8+ T cells in the peripheral lymphoid tissues. The T scm population has been reported as novel memory T cell subgroup (CD44low CD62L+) characterized by high expression amount of stem cell antigen 1 (SCA-1) and capable of self-renew (14-16). This latter characteristic makes the T scm T CD8+ cells a desirable population to develop upon vaccination (14). In line with that, we investigated the expression of SCA-1 in CD44low CD62L+ CD8+T cells in spleens harvested from Mock, VALO-mD901 and PeptiCRAd1 treated groups according to the gating strategy reported in Figure 8—figure supplement 1A we observed that SCA-1 showed a tendency to be upregulated in spleens from mice treated with PeptiCRAd1 (Figure 8—figure supplement 2B), suggesting the possible generation of a long-term CD8+ T cell population in mice upon oncolytic vaccine treatment. No differences were instead observed in CD44+ CD62L+ (T central memory) (Figure 8—figure supplement 2C) or CD44+ CD62L- (T effector memory) (Figure 8—figure supplement 2D) compartments.”.

References

1. Kershaw MH, Hsu C, Mondesire W, Parker LL, Wang G, Overwijk WW, et al. Immunization against endogenous retroviral tumor-associated antigens. Cancer Res. 2001;61(21):7920-4.

2. Casares N, Lasarte JJ, de Cerio AL, Sarobe P, Ruiz M, Melero I, et al. Immunization with a tumor-associated CTL epitope plus a tumor-related or unrelated Th1 helper peptide elicits protective CTL immunity. Eur J Immunol. 2001;31(6):1780-9.

3. McWilliams JA, Sullivan RT, Jordan KR, McMahan RH, Kemmler CB, McDuffie M, et al. Age-dependent tolerance to an endogenous tumor-associated antigen. Vaccine. 2008;26(15):1863-73.

4. Slansky JE, Rattis FM, Boyd LF, Fahmy T, Jaffee EM, Schneck JP, et al. Enhanced antigen specific antitumor immunity with altered peptide ligands that stabilize the MHC-peptide-TCR complex. Immunity. 2000;13(4):529-38.

5. Jordan KR, McMahan RH, Kemmler CB, Kappler JW, Slansky JE. Peptide vaccines prevent tumor growth by activating T cells that respond to native tumor antigens. Proc Natl Acad Sci U S A. 2010;107(10):4652-7.

6. Laumont CM, Vincent K, Hesnard L, Audemard E, Bonneil E, Laverdure JP, et al. Noncoding regions are the main source of targetable tumor-specific antigens. Sci Transl Med. 2018;10(470).

7. Jacopo Chiaro HHEK, Thomas Whalley, Cristian Capasso, Mikaela Grönholm, Sara Feola, Karita D. Peltonen, Firas S. Hamdan, Micaela M. Hernberg, Siru Mäkelä, Hanna Karhapää, Paul E. Brown, Beatriz Martins, Manlio Fusciello, Erkko Ylösmäki, Anna S. Kreutzman, Satu M. Mustjoki ,Barbara Szomolay, Vincenzo Cerullo. Viral Molecular Mimicry Influences the Antitumor Immune Response in Murine and Human Melanoma. medRxiv. 2020.

8. Antunes DA, Rigo MM, Freitas MV, Mendes MFA, Sinigaglia M, Lizee G, et al. Interpreting T-Cell Cross-reactivity through Structure: Implications for TCR-Based Cancer Immunotherapy. Front Immunol. 2017;8:1210.

9. Fluckiger A, Daillere R, Sassi M, Sixt BS, Liu P, Loos F, et al. Cross-reactivity between tumor MHC class I-restricted antigens and an enterococcal bacteriophage. Science. 2020;369(6506):936-42.

10. Larouche JD, Trofimov A, Hesnard L, Ehx G, Zhao Q, Vincent K, et al. Widespread and tissue-specific expression of endogenous retroelements in human somatic tissues. Genome Med. 2020;12(1):40.

11. Chong C, Marino F, Pak H, Racle J, Daniel RT, Muller M, et al. High-throughput and Sensitive Immunopeptidomics Platform Reveals Profound Interferongamma-Mediated Remodeling of the Human Leukocyte Antigen (HLA) Ligandome. Mol Cell Proteomics. 2018;17(3):533-48.

12. Kubiniok P, Marcu A, Bichmann L, Kuchenbecker L, Schuster H, Hamelin DJ, et al. Understanding the Constitutive Presentation of MHC Class I Immunopeptidomes in Primary Tissues. iScience. 2022:103768.

13. Feola S, Capasso C, Fusciello M, Martins B, Tahtinen S, Medeot M, et al. Oncolytic vaccines increase the response to PD-L1 blockade in immunogenic and poorly immunogenic tumors. Oncoimmunology. 2018;7(8):e1457596.

14. Paschall AV, Liu K. An Orthotopic Mouse Model of Spontaneous Breast Cancer Metastasis. J Vis Exp. 2016(114).

15. Gattinoni L, Speiser DE, Lichterfeld M, Bonini C. T memory stem cells in health and disease. Nat Med. 2017;23(1):18-27.

16. Wang Y, Qiu F, Xu Y, Hou X, Zhang Z, Huang L, et al. Stem cell-like memory T cells: The generation and application. J Leukoc Biol. 2021;110(6):1209-23.

17. Zhang Y, Joe G, Hexner E, Zhu J, Emerson SG. Host-reactive CD8+ memory stem cells in graft-versus-host disease. Nat Med. 2005;11(12):1299-305.